# A SIMPLI (Single-cell Identification from MultiPLexed Images) approach for spatially-resolved tissue phenotyping at single-cell resolution

Michele Bortolomeazzi[1,2], Lucia Montorsi[1,2], Damjan Temelkovski[1,2], Mohamed Reda Keddar[1,2], Amelia Acha-Sagredo[1,2], Michael J. Pitcher[3], Gianluca Basso[4,5], Luigi Laghi[4,6], Manuel Rodriguez-Justo[7], Jo Spencer[3] & Francesca D. Ciccarelli[1,2 ✉]

Multiplexed imaging technologies enable the study of biological tissues at single-cell resolution while preserving spatial information. Currently, high-dimension imaging data analysis is technology-specific and requires multiple tools, restricting analytical scalability and result reproducibility. Here we present SIMPLI (Single-cell Identification from MultiPLexed Images), a flexible and technology-agnostic software that unifies all steps of multiplexed imaging data analysis. After raw image processing, SIMPLI performs a spatially resolved, single-cell analysis of the tissue slide as well as cell-independent quantifications of marker expression to investigate features undetectable at the cell level. SIMPLI is highly customisable and can run on desktop computers as well as high-performance computing environments, enabling workflow parallelisation for large datasets. SIMPLI produces multiple tabular and graphical outputs at each step of the analysis. Its containerised implementation and minimum configuration requirements make SIMPLI a portable and reproducible solution for multiplexed imaging data analysis. Software is available at "SIMPLI [https://github.com/ciccalab/SIMPLI]".

[1] Cancer Systems Biology Laboratory, The Francis Crick Institute, London NW1 1AT, UK. [2] School of Cancer and Pharmaceutical Sciences, King's College London, London SE11UL, UK. [3] School of Immunology and Microbial Sciences, King's College London, London SE19RT, UK. [4] Laboratory of Molecular Gastroenterology, IRCCS Humanitas Research Hospital, Rozzano 20089 MI, Italy. [5] Genomic Unit, IRCCS Humanitas Research Hospital, Rozzano 20089 MI, Italy. [6] Department of Medicine and Surgery, University of Parma, Parma 43121 PR, Italy. [7] Department of Histopathology, University College London Cancer Institute, London WC1E 6JJ, UK. ✉email: francesca.ciccarelli@crick.ac.uk

A detailed investigation of tissue composition and function in health and disease requires spatially resolved, single-cell approaches that precisely quantify cell types and states as well as their interactions in situ. Recent technological advances have enabled to stain histological sections with multiple tagged antibodies that are subsequently detected using fluorescence microscopy or mass spectrometry[1]. High-dimensional imaging approaches such as imaging mass cytometry (IMC)[2], multiplexed ion beam imaging (MIBI)[3], co-detection by indexing (CODEX)[4], multiplexed immunofluorescence (mIF, including cycIF)[5] and multiplexed immunohistochemistry (mIHC)[6,7] enable quantification and localisation of cells in sections from formalin-fixed paraffin-embedded (FFPE) tissues, including clinical diagnostic samples. This is of particular value for mapping the tissue-level characteristics of disease conditions and predicting the outcome of therapies that depend on the tissue environment, such as cancer immunotherapy. For example, a recent IMC phenotypic screen of breast cancer subtypes revealed the association between the heterogeneity of somatic mutations and that of the tumour microenvironment[8]. Similarly, a CODEX-based profile of FFPE tissue microarrays from high-risk colorectal cancer patients correlated $PD1^+CD4^+$ T cells with patient survival[9].

The analysis of multiplexed images requires the conversion of pixel intensity data into single-cell data, which can then be characterised phenotypically, quantified comparatively and localised spatially in the tissue. Currently available tools are technology specific and cover only some steps of the whole analytical workflow (Table 1). For example, several computational approaches have been developed to process raw images and extract single-cell data either interactively (Ilastik[10], CellProfiler4[11], CODEX Toolkit[4]) or via command line (imcyto[12], ImcSegmentationPipeline[13]). Distinct sets of tools can then perform cell phenotyping (CellProfiler Analyst[14], Cytomapper[15], Immunocluster[16]) or analyse cell–cell spatial interactions (CytoMap[17], ImaCytE[18], SPIAT[19], neighbouRhood[20]). Similarly, a few tools enable direct pixel-based analysis through pixel classification[10] or quantification of pixel positive areas[11]. Despite such a variety of tools, none of them can perform all of the required analytical steps in a common pipeline. Two exceptions are histoCAT++[21] and QuPath[22], which however have been developed specifically for interactive use and are not well suited for the analysis of large datasets. Moreover, all of these tools rely on ad hoc configuration files and input formats, making the analysis challenging for users with limited computational skills and restricting the scalability, portability and reproducibility in different computing environments.

Here we introduce SIMPLI (Single-cell Identification from MultiPLexed Images), a tool that combines processing of raw images, extraction of single-cell data, and spatially resolved quantification of cell types or functional states into a single pipeline (Table 1). This is achieved through the integration of well-established tools and newly developed scripts into the same workflow, enabling ad hoc configurations of the analysis while ensuring interoperability between its different parts. SIMPLI can be run on desktop computers as well as on high-performance-computing environments, where it can be easily applied to large datasets due to automatic workflow parallelisation. To demonstrate the flexibility of SIMPLI to work with different technologies and experimental conditions, we analyse the phenotypes and spatial distribution of cells in different tissues (human colon, appendix, colorectal cancer) using multiplexed images obtained with distinct technologies (IMC, mIF, CODEX).

## Results

**Overview of the SIMPLI analytical workflow.** SIMPLI performs the analysis of multiplexed imaging data in three steps (Methods, Fig. 1) integrating well-established and newly developed

**Table 1 Features of representative tools for the analysis of multiplexed imaging data.**

| Computational tool | Image processing | Cell segmentation | Cell phenotyping preselected | Cell phenotyping unsupervised | Spatial analysis homotypic | Spatial analysis heterotypic | Pixel analysis | Parallelisation | Imaging technologies |
|---|---|---|---|---|---|---|---|---|---|
| SIMPLI | Yes | Yes | Yes | Yes | Yes | Yes | Yes | Yes | 1-6 |
| CODEX Toolkit[4] | Yes | Yes | Yes | Yes | Yes | Yes | No | No | 3 |
| CellProfiler4[11] | Yes | Yes | Yes | No | No | Partial | Yes | Yes | 1-6 |
| HistoCAT++[21] | Yes | Yes | Yes | Yes | No | Yes | No | No | 1,2,4,5 |
| QuPath[22] | Partial | Yes | Yes | No | No | No | Yes | Yes | 1-6 |
| Cytomapper[15] | No | Yes | No | No | Partial | No | Yes | Yes | 1-5 |
| Ilastik[10] | Yes | Yes | Yes | No | No | Partial | Partial | Yes | 1-6 |
| ImcSegmentationPipeline[13] | Yes | Yes | No | No | No | No | No | Yes | 1 |
| Imcyto[12] | Yes | No | No | No | No | No | No | Yes | 1 |
| SPIAT[19] | No | No | Yes | Yes | Yes | Yes | No | No | 2,5,6 |
| Giotto[47] | No | No | No | Yes | Yes | Yes | No | No | 1-6 |
| ImaCytE[18] | No | No | No | Yes | No | No | No | No | 1 |
| CellProfiler Analyst[14] | No | No | Yes | No | No | No | No | No | 1 |
| Immunocluster[16] | No | No | No | Yes | No | Yes | No | No | 1-5 |
| NeighbouRhood[20] | No | No | No | No | No | Yes | No | No | 1 |
| CytoMAP[17] | No | No | No | No | No | Yes | No | No | 2 |

For each tool, reported are the steps of the analytical workflow that it can perform, whether it can be parallelised and the multiplexed imaging platform it can be applied to (1: IMC; 2: mIF; 3: CODEX; 4: MIBI; 5: mIHC; 6: spatial transcriptomic visualisation). A method was considered compatible with a given imaging technology if this was reported in the original publication or other studies.

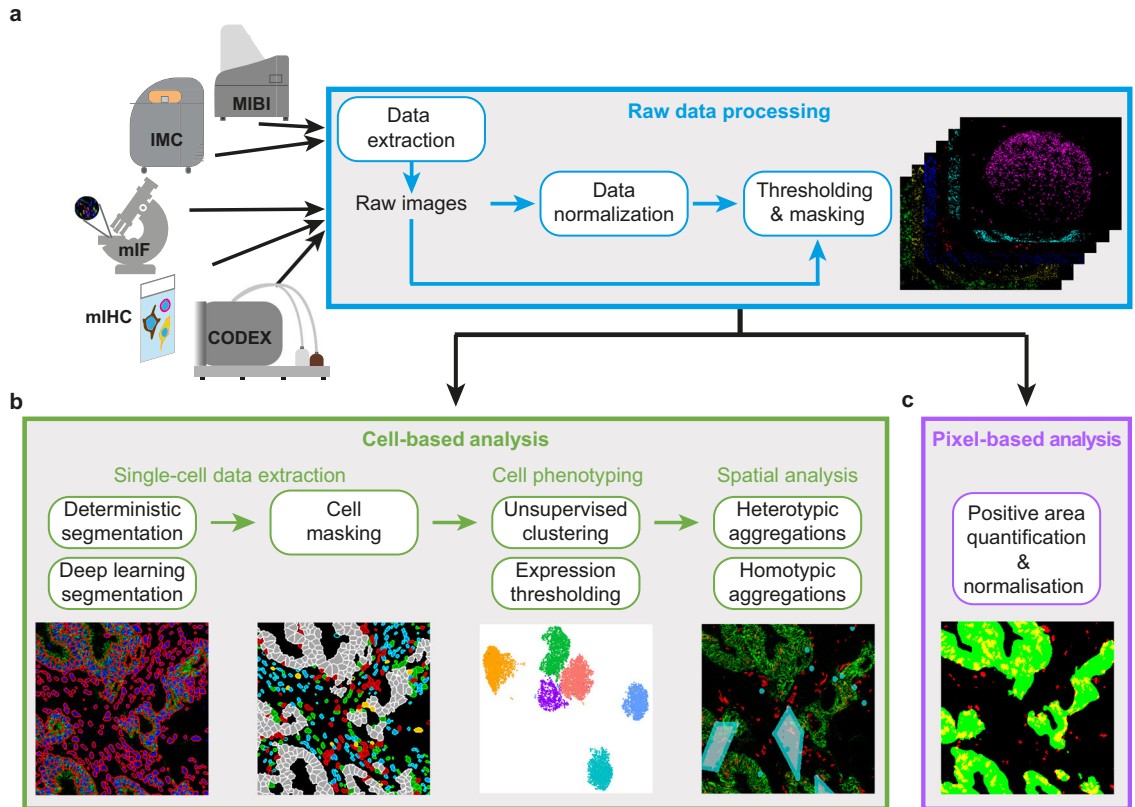

**Fig. 1 Schematics of the SIMPLI workflow. a** Raw images are extracted from IMC or MIBI data or directly imported from other imaging technologies. After their optional normalisation, these images are thresholded to remove the background noise and produce tissue compartments or marker masks. The resulting images can be analysed using a cell-based or a pixel-based approach. **b** In the cell-based analysis, single cells are segmented with deterministic or deep learning models and phenotyped using unsupervised or supervised approaches. The distribution of cells in the tissue can then be investigated through a spatial analysis of homotypic or heterotypic aggregations. **c** In the pixel-based approach, areas positive for a user-defined combination of markers are measured and normalised over the area of the whole image or of the masks defining compartments or areas positive for certain markers.

standalone processes (Supplementary Fig. 1). Each process can be run independently or even skipped with the possibility of using alternative input data at each point of the workflow.

The first step of SIMPLI consists of processing raw data from single or multi-channel images or text files from a variety of high-dimensional imaging technologies (Fig. 1a and Supplementary Fig. 1a). After data extraction, pixel values for each marker can be optionally normalised by rescaling them in each sample. This allows the user to apply the same thresholds for background noise reduction across samples. Alternatively, the normalisation step can be skipped and sample-specific thresholds can be applied directly to individual, non-normalised images to minimise the effect of non-uniform staining. This is recommended for example if markers have low signal-to-noise ratios and the resulting thresholds may be too restrictive or if platform-specific normalisation is required. In the last step of data processing, masks for specific tissue compartments or markers are derived using a fully customisable pipeline based on CellProfiler4[11], where the user can apply filters, thresholds and morphological operations to each image. The resulting processed images can then be analysed at the cell (Fig. 1b) and pixel (Fig. 1c) levels.

The cell-based analysis aims to investigate the qualitative and quantitative cell composition of the tissue and is composed of (1) single-cell data extraction, (2) cell phenotyping and (3) spatial analysis of cell–cell distances (Fig. 1b).

To extract cell data, SIMPLI implements single-cell segmentation using either a deterministic[11] or a deep learning[23] approach (Supplementary Fig. 1b). The former enables deterministic filtering based on cells size and shape, as well as marker

intensities. The latter applies pre-trained models (either provided by SIMPLI or supplied by the user) to identify cells with high accuracy. After cell segmentation, SIMPLI produces the masks of the individual cells and calculates the expression values for each marker in each cell. Cells belonging to tissue compartments or positive for certain markers can then be identified based on their overlap with the previously derived tissue or marker masks.

To define the cell phenotypes, SIMPLI uses two alternative approaches (Supplementary Fig. 1b). The first applies unsupervised clustering to all cells or preselected subsets of cells (for example those mapping to specific tissue compartments or positive for certain markers) using marker expression levels. This leads to the unbiased classification of cells into clusters with similar expression profiles indicating similar phenotypes. The second approach identifies cells with designated phenotypes by applying combinations of user-defined thresholds to the expression values of the markers of interest. These thresholds can be identified through an expert-guided examination of the original images or using the visualisation plots produced by SIMPLI. The two approaches can be used independently or as cross-validation of the cell phenotypes.

To identify cell aggregations within the same (homotypic) or across different (heterotypic) cell types, SIMPLI implements a spatial analysis of the distance between cells within the imaged tissue (Supplementary Fig. 1b). In the case of homotypic aggregations, SIMPLI identifies groups of cells of the same type within a user-defined distance and visually localises them as clusters in the tissue image. In the case of heterotypic aggregations, SIMPLI computes the distance distribution between

distinct cell types and compares them across cell types and experimental conditions. Observed distance distributions can also be compared to expected distributions obtained by randomly reshuffling the cell identities in each sample.

The pixel-based approach implemented in SIMPLI enables quantification of areas positive for a specific marker or combination of markers, independently of their association with cells (Fig. 1c and Supplementary Fig. 1c). The obtained marker-positive areas are then normalised over the area of the whole image, or those of specific tissue compartments or positive for certain markers using the predefined masks, to allow comparisons across samples. The pixel-based analysis is useful for the investigation of tissue features that are not detectable at the cell level. For instance, extracellular or secreted proteins cannot be quantified with approaches dependent on cell segmentation. In addition, being completely cell agnostic, the pixel-based analysis can provide independent validation of cell-based observations.

SIMPLI generates tables, plots and images as outputs of each process, thus enabling the visualisation of results at every step of the analysis.

**IMC quantification of secreted and cell-associated IgA in human colon**. To test its performance and versatility, we applied SIMPLI to four case studies of multiplexed images obtained with different technologies and with diverse origin, size and resolution of the tissue sections (Table 2).

As a first case study, we used SIMPLI to compare the levels of secreted and cell-associated immunoglobulin A (IgA), the major immunoglobulin isotype in intestinal mucosa[24], from IMC-derived multiplexed images of normal human colon. We stained six colon sections (CLN1-CLN6, Supplementary Data 1) with 26 antibodies marking T cells, macrophages, dendritic cells and B cells as well as stromal components (Supplementary Data 2) and ablated one region of interest (ROI) per sample.

Using SIMPLI, we extracted and normalised the 28 single-channel images (26 antibodies and two DNA intercalators) for each of the six ROIs and combined them into a single image per ROI (Fig. 2a). This normalisation enabled the selection of a single threshold for each marker to be used across all samples, thus reducing the complexity of the analysis configuration. By applying these thresholds to the E-cadherin and vimentin expression, we obtained the masks for the epithelium and the lamina propria, respectively (Fig. 2b). We used these masks to assign cells to the two compartments and normalise marker values or positive areas in the downstream analyses.

We then used the pixel-based approach to quantify both the IgA expressed by the plasma cells resident in the diffuse lymphoid tissue of the lamina propria as well as the secreted IgA undergoing transcytosis to traverse the epithelial compartment (Fig. 2b). As expected, most secreted IgA was localised in the epithelial crypts with only minimal presence of IgA$^+$ area in the surface epithelium (Supplementary Fig. 2a). Quantification of the normalised IgA$^+$ areas in the two compartments (Supplementary Fig. 2b) confirmed higher IgA$^+$ levels in the lamina propria than in the epithelium (Fig. 2c). To assess the impact of image normalisation performed in the data processing step, we repeated the same analysis starting from the raw images and applying sample-specific thresholds to remove the background noise. The resulting IgA levels correlated linearly with those obtained from normalised images (Supplementary Fig. 2c), showing that data normalisation does not impact the results.

Next, we quantified the IgA$^+$ plasma cells in the lamina propria using the cell-based approach. First, we performed single-cell segmentation with the deterministic approach and retained only cells overlapping for at least 30% or their area with the lamina

propria mask (Fig. 2d and Supplementary Fig. 2d). We verified that varying the threshold of the overall had a minimal impact on the proportion of cells assigned to the lamina propria (Supplementary Fig. 2e). We then identified IgA$^+$ plasma cells, T cells, macrophages, and dendritic cells resident in the lamina propria according to the highest overlap between the cell area and the mask of each immune cell population (Fig. 2e). Again, we verified that the relative proportion of these cell populations changed only minimally varying the threshold of the overlap with the lamina propria mask (Supplementary Fig. 2f). Finally, we quantified the four immune cell populations across the six samples and observed that IgA$^+$ plasma cells constitute approximately 25% of all immune cells (Fig. 2f). This is consistent with previous quantifications of the fraction of plasma cells over the total mononucleated cells in the lamina propria of healthy individuals[25].

The relative proportion of IgA$^+$ plasma cells positively correlated with the normalised IgA$^+$ area in the lamina propria, demonstrating that the quantification from the single-cell analysis is supported by the cell agnostic measurements at the pixel level (Fig. 2g).

**Localisation of T follicular helper cells in IMC images of a germinal centre**. As a second case study, we used SIMPLI to spatially localise the immune cell populations within a FFPE section of the healthy human appendix (APP1, Supplementary Data 1). After staining the tissue section with 28 markers (26 antibodies and two DNA intercalators, Supplementary Data 2), we performed IMC and used SIMPLI to extract and normalise the single-channel images from the raw IMC data. The resulting combined image revealed a germinal centre in the B cell area and follicle-associated epithelium forming the boundary with the appendiceal lumen (Fig. 3a).

We performed single-cell segmentation with both approaches implemented in SIMPLI and observed high overlap in the identified cells (Supplementary Fig. 3a), indicating a good concordance between the two methods. We then classified 7573 cells obtained with the deterministic segmentation approach in immune and epithelial cells based on the highest overlap with the corresponding masks obtained in the data processing step (Fig. 3b, c). We obtained similar proportions of cells starting from the raw data and applying the z-score normalisation and k-means clustering as implemented in Histocat[26] (Supplementary Fig. 3b), again demonstrating that the normalisation implemented in SIMPLI does not impact the downstream analysis.

Next, we used both methods implemented in SIMPLI to further phenotype the T cells identified within the ROI. First, we applied unsupervised clustering using seven markers of T cell function (Supplementary Data 2). After inspection of the resulting clusters at different resolution levels, we selected 0.25 resolution that returned five distinct cell clusters (Fig. 3d). Based on the marker expression profiles, we assigned cluster 1 to CD4$^+$ T cells, cluster 2 to CD8$^+$CD45RO$^+$ T cells, cluster 3 to CD4$^+$CD45RA$^+$ T cells, cluster 4 to CD4$^+$CD45RO$^+$ T cells and cluster 5 to PD1$^+$CD4$^+$ T cells (Fig. 3e). The latter likely represented a set of PD1$^+$ T follicular helper cells known to be located in the germinal centre[27]. Interestingly, at higher resolution levels, cluster 5 was further divided into two smaller clusters showing PD1 high and low expression (Supplementary Fig. 3c). Similarly, clusters 1 and 2 were further divided into smaller subpopulation based on CD4 and CD45RO expression levels, respectively (Supplementary Fig. 3c). Therefore, although higher resolution levels increase the granularity of cell phenotyping, the unsupervised clustering approach implemented in SIMPLI is robust in identifying similar phenotyping clusters independently of the chosen resolution.

**Table 2 Description of the case studies used to test SIMPLI.**

| Case study | Imaging technology | Analysed samples (n) | Channels (n) | ROI (mm²) | Resolution (µm/pixel) | HPC platform | CPU time (h) | Elapsed real time (h) | RAM (GB) | Processes |
|---|---|---|---|---|---|---|---|---|---|---|
| 1 (Fig. 2) | IMC | 6 | 28 | 1.00 | 1.00 | SGE | 00:20:41 | 00:06:10 | 4.1 | Raw data processing<br>Cell masking<br>Single-cell quantification<br>Pixel intensity comparison |
| 2 (Fig. 3) | IMC | 1 | 28 | 1.00 | 1.00 | SLURM | 00:06:25 | 00:05:30 | 4.2 | Raw data processing<br>Cell masking<br>Unsupervised clustering<br>Expression thresholding<br>Homotypic cell distances |
| 3 (Fig. 4) | mIF | 1 | 7 | 5.45 | 0.50 | SLURM | 00:11:45 | 00:08:23 | 16.7 | Thresholding and masking<br>Expression thresholding<br>Heterotypic cell distances |
| 4 (Fig. 5) | CODEX | 35 | 58 | 1.13 | 0.38 | SGE | 02:32:35 | 00:26:01 | 22.5 | Expression thresholding<br>Heterotypic cell distances |

For each case study, listed are the imaging technologies used to generate the tissue images, the number of samples and markers used, the size of the analysed region of interest (ROI), the resolution of the obtained images, the high-performance (HPC) platform and the computational resources employed to perform the analysis. These include the central processing unit (CPU) time and the elapsed real time, as well as the maximum random access memory (RAM) memory used. Finally, the specific analytical processes performed in each case study are also listed (single-cell segmentation was performed in all of them).
SGE Sun Grid Engine, SLURM Simple Linux Utility for Resource Management.

We re-identified the PD1⁺ T follicular helper cells with the second phenotyping approach based on expression thresholding of CD4 and PD1. Starting from all T cells, we first extracted CD4⁺ T cells (≥0.1 CD4 expression, Fig. 3f) and, within those, we further identified PD1⁺ cells (≥0.15 PD1 expression, Fig. 3g). Both thresholds were chosen after manual inspection of the histological images. The expression profile of the resulting PD1⁺CD4⁺ T cells (Fig. 3h) closely recapitulated that of cluster 5 (Fig. 3e). We repeated the same analysis for clusters 1–4 confirming the high overlap between cells in unsupervised clusters and those re-identified using marker expression thresholds (Supplementary Fig. 3d). Moreover, these cells showed similar expression profiles (Supplementary Fig. 3e) and spatial localisation (Supplementary Fig. 3f), indicating that cell phenotypes identified with unsupervised clustering can be confirmed through user-guided thresholding of marker expression.

Finally, we investigated the spatial localisation of PD1⁺ T follicular helper cells within the ROI by analysing their homotypic aggregations. This allowed us to localise a single high-density cluster containing 84% of PD1⁺CD4⁺ T cells within the germinal centre (Fig. 3i). This distribution of PD1⁺CD4⁺ T cells was in accordance with the localisation of T helper cells in the follicles of secondary lymphoid organs[27] and was confirmed by the histological inspection of the tissue image (Fig. 3j).

**mIF analysis of spatially resolved cell–cell interactions in rectal cancer.** As a third case study, we applied SIMPLI to the spatial analysis of mIF-derived images of a rectal cancer sample (CRC1, Supplementary Data 1) stained with anti CD8, PD1, Ki67, PDL1, CD68, GzB and 4',6-diamidino-2-phenylindole (DAPI) antibodies (Supplementary Data 2). We focused on a 5-mm² ROI, rich in T cells and located at the invasive margin of the tumour (Fig. 4a). This allowed us to characterise the cell–cell interactions between PDL1⁺ cells and PD1⁺CD8⁺ T cells at the tumour boundary in a larger ROI, supporting the scalability of SIMPLI to the analysis of large regions (Table 2).

After image normalisation and single-cell segmentation, we identified PDL1⁺ and PD1⁺CD8⁺ cells by applying expert-defined thresholds to PDL1 (≥0.01), CD8 (≥0.01), and PD1 (≥0.005) expression levels, respectively. We extracted 2026 PDL1⁺ cells (Fig. 4b) and 3177 CD8⁺ cells, 94 of which also expressed PD1 (Fig. 4c). The two sets of PDL1⁺ and PD1⁺CD8⁺ cells constituted 3.7% and 0.2% of all cells in the analysed region, respectively (Fig. 4d). We confirmed similar proportions of PDL1⁺ and PD1⁺CD8⁺ cells by performing signal unmixing, cell segmentation and cell phenotyping with the Inform tissue analysis software[28] (Akoya Biosciences, Fig. 4e).

We characterised the spatial relationship between these cells, focusing on the ones in close proximity to each other. Using the Euclidean distances between their centroids, we identified 35 PDL1⁺ cells and 21 PD1⁺CD8⁺ T cells at a distance lower than 12 µm apart, which corresponded to twice the maximum cell radius length. We considered these cells proximal enough to be engaging in PD1-PDL1 mediated interactions. By comparing PD1⁺CD8⁺ T cells proximal to PDL1⁺ cells and PD1⁺CD8⁺ T cells distal to PDL1⁺ cells, we found no difference in the expression of cytotoxicity (GzB) or proliferation (ki67) markers (Fig. 4f). This is in line with the broad range of cytotoxic activity in this T cell subset observed in colorectal cancer[29]. On the contrary, PDL1⁺ cells proximal to PD1⁺CD8⁺ T cells expressed higher levels of CD68 than PDL1⁺ cells distal to PD1⁺CD8⁺ T cells (Fig. 4g), suggesting spatial proximity between PDL1⁺ macrophages and PD1⁺CD8⁺ T cells. To validate this observation, we identified 1392 macrophages by applying an expert-defined threshold to CD68 expression value (≥0.01, Fig. 4h). We then classified these macrophages as PDL1⁻ and

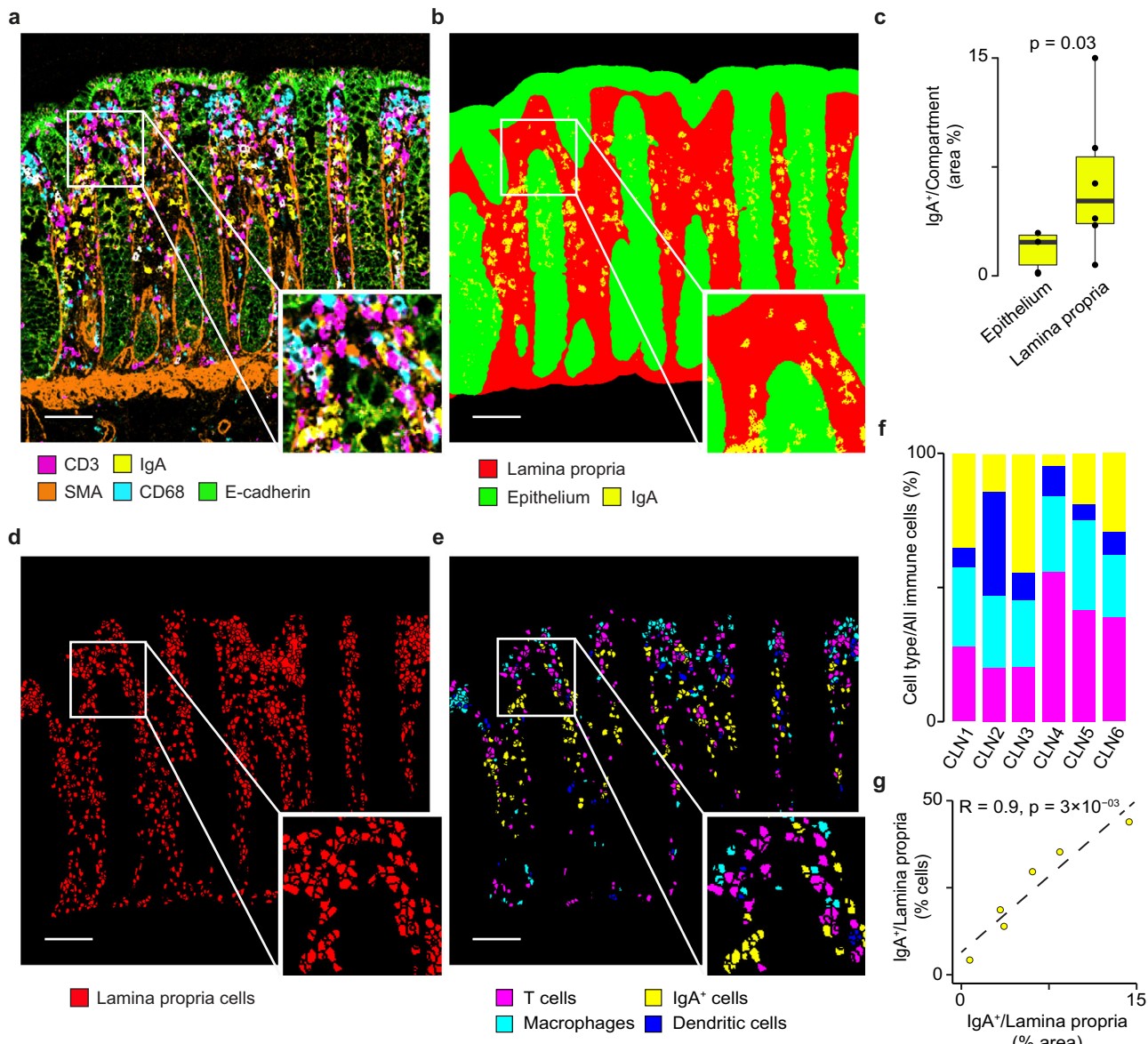

**Fig. 2 IgA quantification in human colon mucosa. a** IMC image of a representative sample (CLN6) of normal colon mucosa after extraction and normalisation of raw data. **b** Masks defining the lamina propria and the epithelial compartments overlaid with IgA$^+$ areas. Lamina propria and epithelial masks were obtained by thresholding the vimentin and E-cadherin channels, respectively. **c** Comparison of normalised IgA$^+$ areas in the lamina propria and epithelial compartments in six independent biological samples (CLN1-CLN6). Normalised areas were measured as the proportion of IgA$^+$ area over the lamina propria and epithelium masks, respectively. Data are presented as a box centred around the median and extending from the first to the third quartile. Whiskers represent the minimum and maximum values. An exact *p* value was calculated using a two-sided Wilcoxon test. **d** Outlines of the cells in the lamina propria. After single-cell segmentation, all cells overlapping with the lamina propria mask by at least 30% of their area were considered as cells resident in the lamina propria. **e** Outlines of immune cells resident in the lamina propria identified according to the highest overlap between their area and the masks for IgA$^+$ cells, T cells, macrophages and dendritic cells. **f** Relative proportions of T cells, IgA$^+$ cells, macrophages and dendritic cells over all immune cells in the lamina propria across CLN1-CLN6. **g** Correlation between normalised IgA$^+$ area and the proportion of IgA$^+$ cells over the total immune cells in the lamina propria in six independent biological samples (CLN1-CLN6). Pearson correlation coefficient R and associated *p* value based on Fisher's Z transform are shown. Images in panels (**a**), (**b**), (**d**), (**e**) were derived from a representative sample (CLN6, Supplementary Data 1). CD3 and T cells, magenta; IgA and IgA$^+$ cells, yellow; Smooth Muscle Actin (SMA), orange; CD68 and macrophages, cyan; E-cadherin and epithelial cells, green; Lamina propria and lamina propria cells, red; Dendritic cells, blue. Scale bar in all images = 100 μm. Source data are provided as a Source Data file.

PDL1$^+$ cells, respectively, using 0.1 PDL1 expression threshold. Comparing the distance of the resulting two populations from the nearest PD1$^+$CD8$^+$ T cells, we confirmed that PDL1$^+$CD68$^+$ macrophages were significantly closer to PD1$^+$CD8$^+$ T cells than PDL1$^-$CD68$^+$ macrophages (Fig. 4i). By inspecting the imaged tissue at ×40 magnification, we confirmed the localisation of PDL1$^+$CD68$^+$ macrophages in close proximity to PD1$^+$CD8$^+$

cells, as well as the presence of both PD1$^+$CD8$^+$GzB$^-$ T cells and PD1$^+$CD8$^+$GzB$^+$ T cells proximal to PDL1$^+$ cells (Fig. 4j).

**Comparison of cell distances in CODEX images of colorectal cancer subtypes.** As a fourth case study, we used SIMPLI to compare the distances between immune cells and tumour or

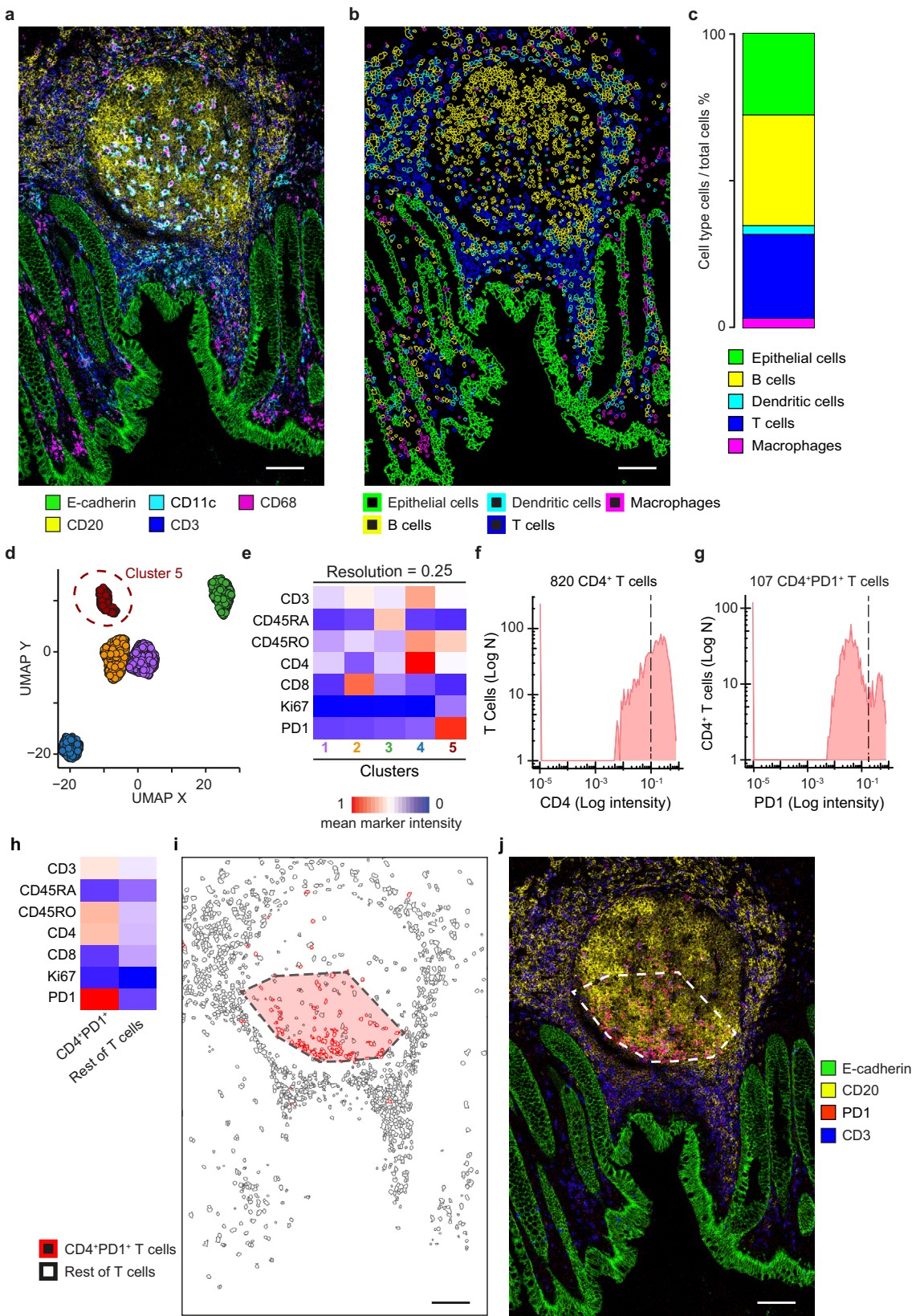

endothelial cells in Crohn's-like reaction (CLR) and diffuse inflammatory infiltration (DII) colorectal cancer subtypes[9]. The high-dimensional imaging data were derived from 35 colorectal cancer samples (17 CLRs and 18 DIIs, Supplementary Data 1) and were obtained using CODEX with a 56 marker panel[9]

(Supplementary Data 2). Such a large number of antibodies enabled the identification and spatial localisation of T cells, B cells, plasma cells, macrophages, NK cells, granulocytes, dendritic cells, tumour cells, neuroendocrine cells, smooth muscle, nerves, lymphatic and blood vessels (Fig. 5a).

**Fig. 3 Single-cell characterisation of T cells in a human germinal centre. a** IMC image of a normal appendix (APP1) showing a central germinal centre with the columnar epithelium delimiting the appendiceal lumen. **b** Outlines of T cells, B cells, macrophages, dendritic and epithelial cells identified through the highest overlap with the respective masks. **c** Proportions of T cells, B cells, macrophages, dendritic and epithelial cells over all cells. **d** UMAP plot of 1466 T cells grouped in five clusters resulting from unsupervised clustering according to the expression of seven markers of T cell function (Supplementary Data 2). Cluster 5 (circled) corresponds to PD1+CD4+ T cells. **e** Expression profiles of the five clusters identified in (**d**). The mean intensity value of each marker across all cells is reported. The colour scale was normalised across all markers and cells. **f** Density plots of CD4 expression in T Cells. Cells with ≥0.1 CD4 expression were considered as CD4+ T cells. **g** Density plot of PD1 expression in CD4+ T cells. Cells with ≥0.15 PD1+ expression were considered as PD1+CD4+ T cells. Thresholds for CD4 and PDL1 were identified through histological inspection of the PD1 channel images. **h** Expression profiles of the PD1+CD4+ T cells and the rest of T cells. For both populations, the mean intensity value of each marker across all cells is shown. The colour scale was normalised across all markers and cells. **i** Position map of T cells within the ROI. The area of a high-density cluster of ≥5 PD1+CD4+ T cells per 10,000 $\mu m^2$ is highlighted in red. **j** IMC image showing the localisation of the PD1 signal within the ROI. Images in (**a**), (**b**), (**i**), and (**j**) were derived from a single experiment (APP1, Supplementary Data 1). Panels (**a**), (**b**), (**c**), (**i**), and (**j**): E-cadherin and epithelial cells, green; CD11c and Dendritic cells, cyan; CD68 and macrophages, magenta; CD20 and B cells, yellow; CD3 and T cells, blue; PD1 and PD1+CD4+ cells, red. Panels (**d**) and (**e**): cluster 1, violet; cluster 2, orange; cluster 3, green; cluster 4, blue; cluster 5, red. Scale bar for all images = 100 μm. Source data are provided as a Source Data file.

We processed the raw data from the original study, including normalisation. We then performed single-cell segmentation and quantified the main cell types identified in the original study[9] by applying expert-defined thresholds to the expression of markers representative of each population (CDX2, MUC1 or cytokeratin for tumour cells; CD34 or CD31 for endothelial cells; vimentin for stromal cells; CD11c for dendritic cells; CD38 for B cells; CD3 and CD4 for CD4+ T cells; CD3, CD4 and FOXP3 for Tregs; CD3 and CD8 for CD8+ T cells, CD68 for macrophages). The obtained relative proportions of immune cells across all samples were highly concordant with those reported in the original study[9] (Fig. 5b).

We then measured the distances of the main immune cell types from tumour cells and blood vessels by performing a heterotypic spatial analysis as implemented in SIMPLI. First, we calculated the distances of each macrophage, CD8+ T cell, CD4+ T cell, Treg and B cell to the nearest tumour cell or endothelial cell using the coordinates of the cell centroids. From these, we derived the corresponding distance distributions from the nearest tumour cell or endothelial cell in each sample. Finally, we compared the resulting distributions between 17 CLR and 18 DII colorectal cancer subtypes. After correcting for multiple testing, we considered biologically relevant only differences between the median distances of the two sample subtypes bigger than 8 μm, corresponding to the diameter of B and T lymphocytes[30]. With this approach, we found that Tregs were significantly closer to tumour cells in DII (median distance = 22.4 μm) compared to CLR (35.6 μm, Fig. 5c). On the contrary, B cells were more proximal to blood vessels in CLR (33.5 μm) than in DII (43.3 μm, Fig. 5d). We further supported these results with a permutation test, where we re-labelled randomly the identities to all cells in each sample for 10,000 times to derive an expected distribution of differences in distances between CLR and DII cells. The comparisons of observed values to the expected distributions, confirmed that Tregs were significantly closer to tumour cells in DII (Fig.5e) while B cells were significantly closer to blood vessels in CLR (Fig. 5f). Since the spatial randomness used as a baseline for the permutation test is an approximation of the highly organised structure of biological tissues, we sought further support this result through independent inspection of the spatial distributions of B cells in CLRs (Fig. 5g) and DII (Fig. 5h) in the histological images.

This result, not reported in the original study, showcases the discovery potential of the quantitative analysis of spatial relationships between cell populations implemented in SIMPLI. In addition, the SIMPLI graphical representations of the tissue composition as an overlay of cell boundaries colour-coded by cell populations greatly facilitate the visual inspection of their spatial interactions in their original tissue context.

## Discussion

SIMPLI is an open-source, customisable and technology-independent tool for the analysis of multiplexed imaging data. It enables the processing of raw images, the extraction of cell data and the spatially resolved quantification of cell types or functional states as well as a cell-independent analysis of tissues at the pixel level, all within a single platform (Table 1). Moreover, it gives high flexibility to the user who can decide whether to skip processes implemented in SIMPLI and replace them with external tools to then re-start the pipeline at any point.

In comparison to currently available software, SIMPLI increases the portability, scalability and reproducibility of the analysis (Table 2). Moreover, it can easily accommodate specific analytical requirements across a wide range of tissues and imaging technologies at different levels of resolution and multiplexing through user-friendly configuration files. SIMPLI interoperates with multiple software and programming languages by leveraging workflow management and containerisation. This makes the inclusion of additional algorithms, features and imaging data formats easy to implement. For example, as possible future developments, SIMPLI may include alternative methods of cell segmentation, pixel and cell classification or a Graphical User Interface for interactive data visualisation. For this reason, we will maintain SIMPLI and its documentation up-to-date and will further expand it to leverage new tools as they become adopted by the community. Similarly, feedback from users will be collected through the dedicated GitHub repository.

Multiplexed imaging methods have proven to be a powerful approach for the study of tissues through the in-depth characterisation of cell phenotypes and interactions. SIMPLI, which was recently able to reveal differences in the composition of the micro-environment between colorectal cancers responsive and resistant to anti-PD1 immunotherapy[31], represents an effort to make these analyses more accessible to a wider community. This will enable the exploitation of highly multiplexed imaging technologies for multiple applications, ranging from basic life science and pharmaceutical research to precision medical use in the clinics.

## Methods

All patients enrolled in this study provided written informed consent in accordance with approved institutional guidelines (University College London Hospital, REC Reference: 20/YH/0088; Istituto Clinico Humanitas, REC Reference: ICH-25-09).

**SIMPLI description and implementation**. SIMPLI's workflow is divided into three steps (raw image processing; cell-based analysis; pixel-based analysis), which are constituted of multiple standalone processes (Fig. 1 and Supplementary Fig. 1). Processes can be executed sequentially or independently from the command line or through a configuration file that can be edited with any text editor. This allows the user to skip some of them and use alternative input data for downstream analyses. In addition, parameters and options can be specified through the same

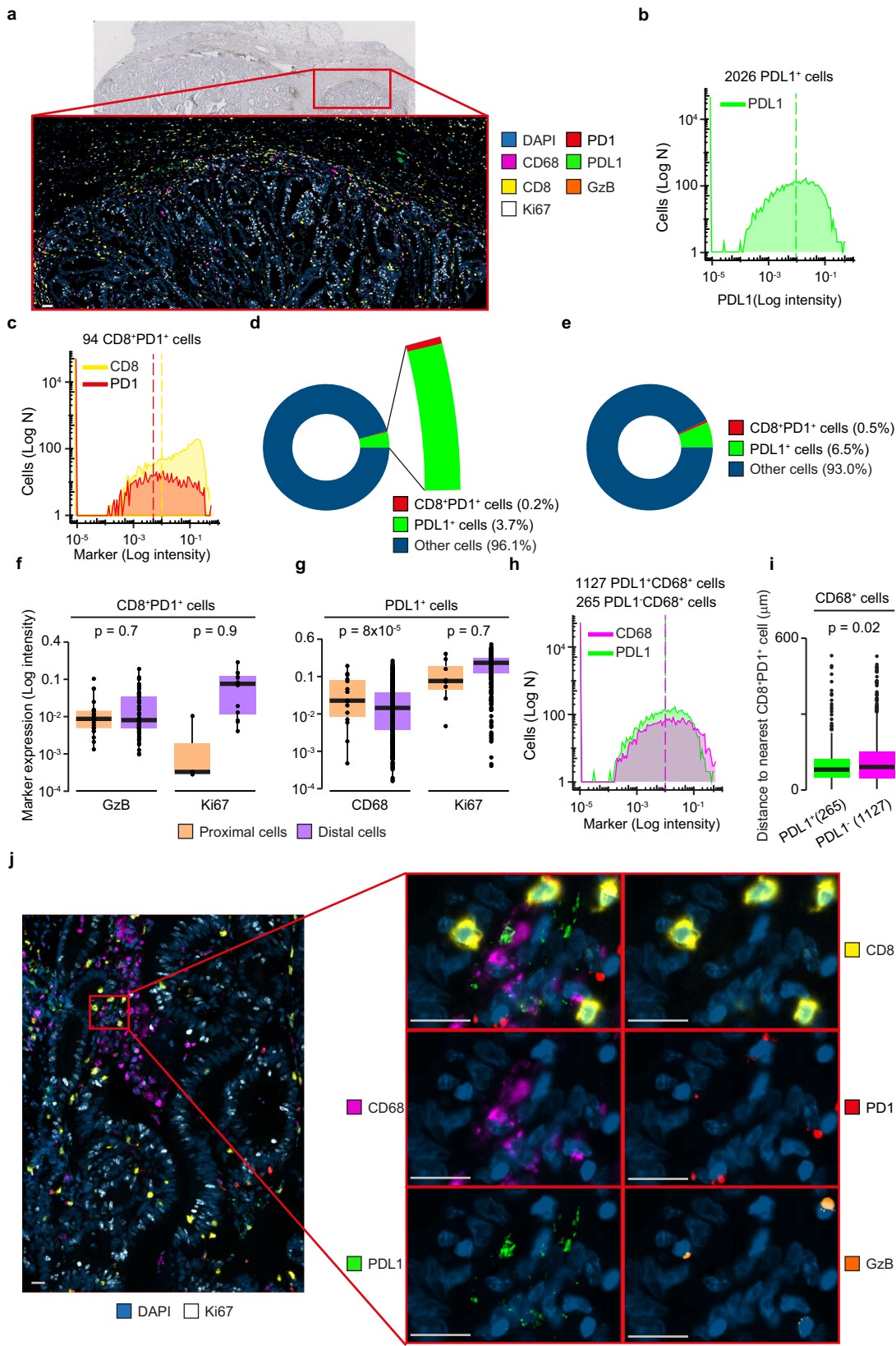

configuration files without the need to set up tool-specific input files in any specific directory structure.

Raw data from IMC or MIBI experiments (.mcd or.txt files) are converted into single or multi-channel.tiff images with imctools[32]. Data from other multiplexed imaging platforms may be supplied directly as raw single or multi-channel tiff images (Supplementary Fig. 1a). Raw images can be thresholded individually to minimise the effect of non-uniform staining and then used directly for the cell- and

pixel-based analyses. Alternatively, they can be first normalised across samples by rescaling pixel values of each channel up to the 99th percentile of the distribution using the EBImage[33] package and custom R scripts. Normalised images can then be processed with CellProfiler4[11] to generate thresholded images and masks of tissue compartments or markers to be used in the following steps. In this step, the user can apply a range of filters, thresholds and morphological operations to each image, according to the experimental plan.

**Fig. 4 Characterisation of PDL1[+] and PD1[+] cells at the tumour invasive margin. a** CD3 immunohistochemistry (main image) and sequential mIF image (zoom-in, ×20 magnification) of a rectal cancer sample (CRC1). The mIF image corresponded to a 5 mm² tissue area at the invasive margin of the tumour and was obtained by combining the pre-processed images of seven markers. Scale bar = 50 μm. **b** Density plot of PDL1 expression in CD8⁻ cells. Cells with ≥0.01 PDL1 expression were considered as PDL1[+] cells. **c** Density plots of CD8 and PD1 expression in T cells. Cells with ≥0.01 CD8 expression and ≥0.005 PD1 expression were considered as PD1[+]CD8[+] T cells. Expression thresholds were identified through histological inspection of PDL1, CD8 and PD1 channel images and are indicated as dotted lines in the corresponding plots. **d** Proportions of PD1[+]CD8[+] cells and PDL1[+] cells over total cells, as measured using SIMPLI data processing, including normalisation. **e** Proportions of PD1[+]CD8[+] cells and PDL1[+] cells over total cells, as measured using the Inform tissue analysis software package[28]. **f** Comparison of the mean intensity of GzB and Ki67 between PD1[+]CD8[+] T cells proximal (n = 21) and distal (n = 73) to PDL1[+]cells. Proximal PD1[+]CD8[+] T cells were defined as those at less than 12 μm from a PDL1[+] cell. **g** Comparison of the mean intensity of CD68 and Ki67 between PDL1[+]cells proximal (n = 35) and distal (n = 1991) to PD1[+]CD8[+] T cells. Proximal PDL1[+] cells were defined as those at less than 12 μm from a PD1[+]CD8[+] T cell. **h** Density plots of CD68 and PD1 expression in all cells. Cells with ≥0.01 CD68 and PDL1 expression were considered as PDL1[+]CD68[+] cells. **i** Comparison of distance of PDL1[+] (n = 265) and PDL1⁻ (n = 1127) CD68[+] macrophages to the nearest PD1[+]CD8[+] T cell. Data in (**f**), (**g**) and (**i**) are presented as a box centred around the median and extending from the first to the third quartile. Whiskers represent the minimum and maximum values. An exact p value was calculated using a two-sided Wilcoxon test. **j** High-resolution (×40 magnification) mIF image of PD1[+]CD8[+] T cells proximal to PDL1[+]CD68[+] cells. Zoom in images show each marker separately and merged. Scale bar = 20 μm. Images in (**a**) and (**j**) were derived from a single experiment (CRC1, Supplementary Data 1). DAPI and other cells, blue; PD1 and PD1[+]CD8[+] T cells, red; CD68 and PDL1⁻ CD68[+] cells, magenta; PDL1 and PDL1[+]CD68[+] cells, green; CD8, yellow; Granzyme B (GzB), orange; Ki67, white; proximal cells, pink; distal cells, violet. Source data are provided as a Source Data file.

Pixel-based and cell-based analyses can be run as single workflows or in parallel within the same run. Both of them provide multiple outputs of the various processes, including tabular text files, visualisation plots and comparisons across samples (Supplementary Fig. 1).

The cell-based analysis is composed of cell data extraction, cell phenotyping and spatial analysis (Supplementary Fig. 1b). The extraction of cell data starts with single-cell segmentation using CellProfiler4[11] or StarDist[23] with scikit-image[34] used for feature extraction. In the latter case, default models or user-provided trained models can be used. Cell segmentation returns (1) single-cell data consisting of the marker expression values and the coordinates of each cell in the ROI and (2) the ROI segmentation mask marking all the pixels belonging to each cell with its unique identifier. Cells mapping to tissue compartments or positive for certain markers can then be identified based on their overlap with the tissue compartments or marker masks derived in the previous step. These cells are visualised in the ROI as outlines, while their proportions are quantified in barplots and boxplots.

All cells or only those in specific tissue compartments or positive for certain markers can be further phenotyped using two approaches. The first consists of unsupervised clustering based on the marker expression values using Seurat[35]. Cells are represented as nodes in a k-nearest neighbour graph based on their Euclidean distances in a principal component analysis space. This graph is then partitioned into clusters using the Louvain algorithm[36] at user-defined levels of resolution leading to the unsupervised identification of cell phenotypes. Clusters of cell phenotypes are plotted as scatterplots in Uniform Manifold Approximation and Projection (UMAP)[37] space. The second phenotyping approach is based on user-defined thresholds of marker expression values that can be combined using logical operators for the identification of designated cell phenotypes. The distributions of cells are represented as density plots based on the marker expression levels. In both phenotyping approaches, the expression profiles of the cell types are plotted as heatmaps, their proportions quantified in barplots and boxplots and their locations in the ROI visualised as cell outlines.

Once cell populations and phenotypes have been identified, the spatial analysis investigates the distance between cells of the same (homotypic aggregations) or different (heterotypic aggregations) types. The homotypic and heterotypic spatial analyses can be run in parallel or singularly on one or more sets of cells. In the homotypic analysis, clusters of cells of the same type within a user-defined distance are identified with DBSCAN[38] as implemented in the fpc[39] R package. These homotypic cell aggregations are visualised as position maps, reporting cell location and high-density clusters in the ROI. In the heterotypic analysis, the cell distances, defined as the Euclidean distances between cell centroids, are computed using custom R scripts and visualised as density plots. The resulting distribution of cell distances can be compared between group of samples using a two-sided Wilcoxon test with Benjamini–Hochberg FDR correction. Observed distances can also be compared to the distribution of expected distances obtained by reshuffling cell identities in each sample randomly for a user-defined number of times (default value = 10,000 reshufflings, Supplementary Fig. 1b). The statistical significance of this comparison is evaluated with a two-tailed permutation test adjusted for multiple hypothesis testing with the Benjamini–Hochberg correction.

The pixel-based analysis quantifies areas positive for a user-defined combination of markers using the EBImage[33] package with custom R scripts (Supplementary Fig. 1c). These measurements are performed starting from the thresholded images produced in the raw image processing step (Supplementary Fig. 1a). The marker-positive areas obtained in this way are then normalised over the area of the whole image or specific tissue or marker compartments. The resulting normalised positive areas can then be quantified in barplots and boxplots.

SIMPLI is implemented as a Nextflow[40] pipeline employing Singularity containers[41] hosted on Singularity Hub[42] to manage all the libraries and software tools. This allows SIMPLI to automatically manage all dependencies, irrespective of the running platform. Nextflow also manages automatic parallelisation of all processes while still allowing the selection of parts of the analysis to execute.

**Sample description**. Six FFPE blocks of normal (non-cancerous) colon mucosa (CLN1-CLN6), one of normal appendix (APP1) and one of rectal cancer (CRC1) were obtained from eight individuals who underwent surgery for the removal of colorectal cancers (Supplementary Data 1). All blocks were reviewed by an expert pathologist (M.R.-J.).

**Staining and IMC ablation of human colon mucosa and appendix**. Four-μm-thick sections were cut from each block of samples CLN1-CLN6 and APP1 with a microtome and used for staining with a panel of 26 antibodies targeting the main immune, stromal and epithelial cell populations of the gastrointestinal tract (Supplementary Data 2). The optimal dilution of each antibody in the panel was identified by staining and ablating FFPE appendix sections. The resulting images were reviewed by a mucosal immunologist (J.S.) and the dilution giving the best signal to background ratio was selected for each antibody (Supplementary Data 2). To perform the staining for IMC, slides were dewaxed after a 1-h incubation at 60 °C, rehydrated and heat-induced antigen retrieval was performed with a pressure cooker in Antigen Retrieval Reagent-Basic (R&D Systems). Slides were incubated in a 10% BSA (Sigma), 0.1% Tween (Sigma), and 2% Kiovig (Shire Pharmaceuticals) Superblock Blocking Buffer (Thermo Fisher) blocking solution at room temperature for 2 h. Each antibody was added to a primary antibody mix at the selected concentration in blocking solution and incubated overnight at 4 °C. After two washes in PBS and PBS-0.1% Tween, the slides were treated with DNA intercalator Cell-ID™ Intercalator-Ir (Fluidigm) (containing the two iridium isotopes 191Ir and 193Ir) 1.25 mM in a PBS solution. After a 30-min incubation, the slides were washed once in PBS and once in MilliQ water and air-dried. The stained slides were then loaded in the Hyperion Imaging System (Fluidigm) imaging module to obtain light-contrast high-resolution images of approximately 4 mm². These images were used to select the ROI in each slide. For CLN1-CLN6, 1 mm² ROIs were selected to contain the full thickness of the colon mucosa, with epithelial crypts in a longitudinal orientation. For APP1, a 1-mm² ROI containing a lymphoid follicle in its whole depth alongside a portion of lamina propria and of the epithelium was selected. ROIs were ablated at a 1 μm/pixel resolution and 200 Hz frequency.

**IMC data analysis of human colon mucosa**. Twenty-eight images from 26 antibodies (Supplementary Data 2) and two DNA intercalators were obtained from the raw.txt files of the ablated regions in CLN1-CLN6 using the data extraction process. Pixel intensities for each channel were normalised to the 99th percentile in all samples and Otsu thresholding was performed on the normalised images with a custom CellProfiler4 pipeline, which was employed also to generate the masks for the lamina propria (using the Vimentin channel including all <75-pixel large negative areas) and the epithelium (starting from the Pan-keratin and E-cadherin channels, dilatating the images with a three-pixel disk and the filling up of all <75-pixel large negative areas). These masks were then added into a sum image, which underwent dilatation with a three-pixel disk and filling up of all <25-pixel large negative areas. Positive features outside of the lamina and epithelium were removed with an opening operation using a 150-pixel radius and the lamina propria mask

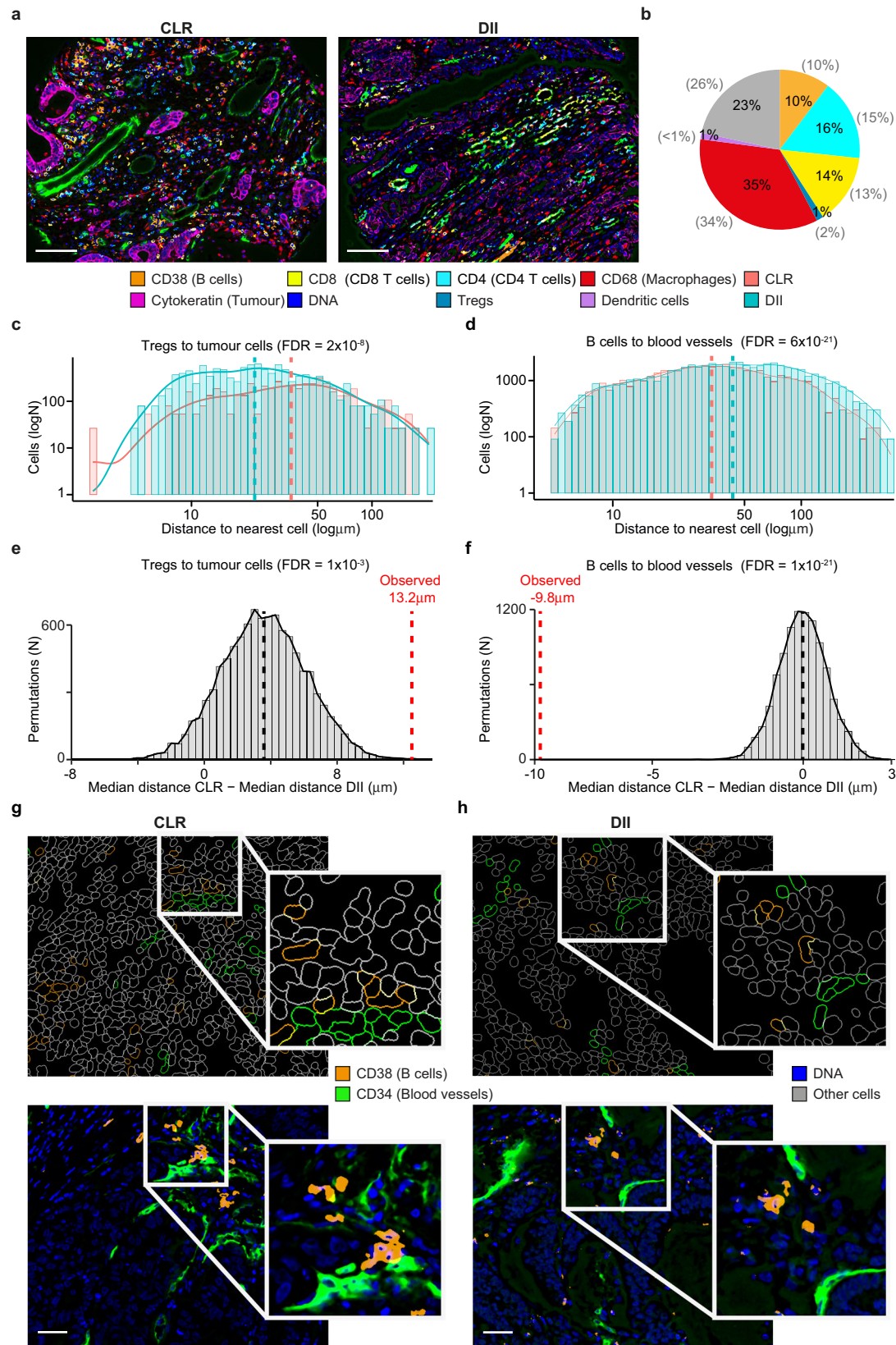

was subtracted from the sum image to generate the final mask for the epithelial compartment. These masks and the thresholded images were used as input for the pixel-based and cell-based analysis processes. The IgA masks employed for the pixel analysis were generated using a three-class global Otsu thresholding with two background classes after applying a Gaussian filter with a 1.5-pixel large radius to remove high-intensity artefacts of that size, which we noticed after manual inspection of the images.

To evaluate the effect of normalisation on the downstream analysis, sample-specific thresholds were manually selected for IgA, E-Cadherin, Pan-Keratin and Vimentin and applied to the raw images. The resulting thresholded images were used to generate lamina propria and epithelial masks for each sample individually.

Pixel-level analysis was performed on the IgA masks derived from either the normalised or the raw images and IgA+ areas in the tissue, lamina propria and epithelium were measured and normalised over the areas of the three compartments.

**Fig. 5 Spatial localisation of immune cells in two colorectal cancer subtypes. a** CODEX images of two representative CLR (CRC_12_24) and DII (CRC_31_16) colorectal cancer samples. **b** Proportions of CD8[+] T cells, CD4[+] T cells, Tregs, macrophages, dendritic cells, B cells and other mixed immune cell populations across the 35 analysed samples. Cell types were identified by applying expert-defined thresholds to the expression intensity of representative markers and normalised over the total non-cancer cells. These thresholds were derived through histological inspection of the channel images. The cell proportion corresponding to each population from the original study[9] is reported in brackets. Distance distribution of Tregs to the nearest tumour cell (**c**) and of B cells to the nearest endothelial cell (**d**) of CLR and DII samples. Distances between cell pairs were calculated using the cell centroids coordinates and the resulting distributions were compared between CRC subtypes using a two-sided Wilcoxon test. Benjamini–Hochberg FDR correction was applied for testing over ten cell type comparisons. Only differences of at least 8 μm and with FDR < 0.1 were considered significant. Dashed lines represent the medians of the distributions. Distribution of the expected differences between the median distances of Tregs to the nearest tumour cell (**e**) and of B cells to the nearest endothelial cell (**f**) in CLR and DII samples. Expected values were calculated with a permutation test, where cell identities were randomly reassigned for 10,000 times within each sample. The resulting median values were compared to the observed differences with a two-tailed permutation test adjusted for multiple hypothesis testing with the Benjamini–Hochberg correction. Single-cell outlines of B cells and blood vessels (upper panel) and associated images (lower panel) form a representative CLR (CRC_17_34) (**g**) and DII (CRC_15_29) (**h**) sample out of 35 colorectal cancer samples (Supplementary Data 1). CD38 and B cells, orange; CD8 and CD8[+] T cells, yellow; CD4 and CD4[+] T cells, cyan; CD68 and macrophages, red; cytokeratin and tumour cells, magenta; DNA, blue; Tregs, teal; dendritic cells, violet. Crohn's-like reaction (CLR) orange; diffuse inflammatory infiltration (DII), teal. Scale bar = 100 μm. Source data are provided as a Source Data file.

Cell-level analysis started with CellProfiler4 segmentation first on DNA1 with global Otsu thresholding to identify the cell nuclei. Then, cells were identified by radially expanding each nucleus for up to 10 pixels over a membrane mask derived from the IgA, CD3, CD68, CD11c and E-cadherin channels. After inspection by an expert histologist (J.S.), only cells overlapping with the lamina propria mask by at least 30% were retained.

Cell identities were assigned according to the highest overlap of the cell area with marker-specific thresholds defined by an expert histologist (J.S.): ≥15% of the IgA mask for IgA cells; ≥15% of the CD3 mask for T cells; ≥25% of the CD68 mask for macrophages; ≥15% of CD11c mask for dendritic cells.

**IMC data analysis of human appendix**. Images from the same 26 antibodies and two DNA intercalators used in the colon mucosa (Supplementary Data 2) were obtained from the raw.txt files of the ablated region in APP1, normalised to the 99th percentile and thresholded with CellProfiler4 as described above. For the cell-based analysis, nuclei were identified using the DNA1 channel and cells were isolated through watershed segmentation with the nuclei as seeds on a membrane mask summing up CD45, Pan-keratin and E-cadherin thresholded images.

Cells were assigned to the epithelium or to immune cell populations if they overlapped for ≥10% with the following masks: CD3 mask for T cells; CD20 and CD27 masks for B cells; CD68 mask for macrophages; CD11c mask for dendritic cells; E-cadherin[+] and Pan-keratin[+] masks for epithelial cells.

T cells were further phenotyped using unsupervised clustering at resolutions between 0.1 and 1.0, with 0.05 intervals and based on the cell marker intensity for CD3, CD45RA, CD45RO, CD4, CD8, Ki67 and PD1. The resulting clusters were manually inspected and the clustering with the highest number of biologically meaningful clusters (resolution = 0.25) was chosen. Clusters were re-identified using mean intensity thresholds defined by an expert histologist (J.S.) for the following markers: CD3 >0.06 for cluster 1; CD8a >0.125 for cluster 2; CD45RA >0.125 for cluster 3; CD4 >0.125 and CD45RO >0.15 for cluster 4; and CD4 >0.1 and PD1 >0.15 for cluster 5.

Homotypic aggregations of PD1[+]CD4[+] T cells (cluster 5, resolution = 0.25) were computed using a minimum of five points per cluster and a reachability parameter corresponding to a density of at least 5 cells/mm[2].

**CD3 staining and mIF of human rectal cancer**. Two 4-μm-thick serial sections were cut from CRC1 FFPE block using a microtome. The first slide was dewaxed and rehydrated before carrying out HIER with Antigen Retrieval Reagent-Basic (R&D Systems). The tissue was then blocked and incubated with the anti-CD3 antibody (Dako, Supplementary Data 2) followed by horseradish peroxidase (HRP) conjugated anti-rabbit antibody (Dako) and stained with 3,3' diaminobenzidine substrate (Abcam) and haematoxylin. Areas with CD3[+] infiltration in the proximity of the tumour invasive margin were identified by a clinical pathologist (M.R.-J.).

The second slide was stained with a panel of six antibodies (CD8, PD1, Ki67, PDL1, CD68, GzB, Supplementary Data 2), Opal fluorophores and DAPI on a Ventana Discovery Ultra automated staining platform (Roche). Expected expression and cellular localisation of each marker as well as fluorophore brightness were used to minimise fluorescence spillage upon antibody-Opal pairing. Following a 1-h incubation at 60 °C, the slide was subjected to an automated staining protocol on an autostainer. The protocol involved deparaffinisation (EZ-Prep solution, Roche), HIER (DISC. CC1 solution, Roche) and seven sequential rounds of 1-h incubation with the primary antibody, 12 min incubation with the HRP-conjugated secondary antibody (DISC. Omnimap anti-Ms HRP RUO or DISC. Omnimap anti-Rb HRP RUO, Roche) and 16-min

incubation with the Opal reactive fluorophore (Akoya Biosciences). For the last round of staining, the slide was incubated with Opal TSA-DIG reagent (Akoya Biosciences) for 12 min followed by Opal 780 reactive fluorophore for 1 h (Akoya Biosciences). A denaturation step (100 °C for 8 min) was introduced between each staining round in order to remove the primary and secondary antibodies from the previous cycle without disrupting the fluorescent signal. The slide was counterstained with DAPI (Akoya Biosciences) and coverslipped using ProLong Gold antifade mounting media (Thermo Fisher Scientific). The Vectra Polaris automated quantitative pathology imaging system (Akoya Biosciences) was used to scan the labelled slide. Six fields of view, within the area selected by the pathologist, were scanned at ×20 and ×40 magnification using appropriate exposure times and loaded into inForm[28] for spectral unmixing and autofluorescence isolation using the spectral libraries.

**mIF data analysis**. After spectral unmixing and merging of six ×20 fields of view for a total of >5 mm[2] ROI (Table 2), one single-tiff image was extracted for each marker and its intensity was rescaled from 0 to 1 with custom R scripts. The resulting single-tiff images were pre-processed to remove the background noise with Otsu thresholding in CellProfiler4 and used for cell segmentation by applying a global threshold to the DAPI channel and selecting all objects with a diameter between four and 60 pixels. PD1[+]CD8[+] cells, CD68[+] cells and PDL1[+] cells were then identified using mean intensity thresholds of 0.01 for CD8, 0.005 for PD1, 0.01 for CD68 and 0.01 for PDL1. All thresholds were inspected by an expert histologist (J.S.). To crosscheck these results, images were analysed with the Inform[28] package. After spectral unmixing, images were segmented with the Adaptive Cell Segmentation option applied to the DAPI channel for nuclei identification ("relative intensity" = 0.1, "splitting sensitivity" = 0.1, "minimum size" = 5). Then PD1[+]CD8[+] cells and PDL1[+] cells were identified.

The distributions of minimum distances between PDL1[+] cells and PD1[+]CD8[+] cells were calculated from the coordinates of the centroids of each cell in the image. All PDL1[+] cells and PD1[+]CD8[+] cells at a distance from each other lower than double the maximum cell radius (24 pixels = 12 μm) were considered as proximal. All other cells were classified as distal.

**CODEX data analysis**. A published dataset of colorectal CODEX images[9] was downloaded from The Cancer Imaging Archive (https://doi.org/10.7937/tcia.2020.fqn0-0326). It consisted of processed CODEX data from 35 colorectal cancer samples divided in two groups (CLR and DII) according to the peritumoral inflammatory levels and the presence of tertiary lymphoid structures[9]. For each sample, four.tiff images were available representing four 0.6-mm spots from two 70-core tissue microarrays. These images were hyperstacks of 58 channels including 56 antibodies (Supplementary Data 2) and two DNA markers with a resolution of 377 nm/pixel. After the manual review of all 140 spots, one representative image per sample was selected, having the best focus and containing both tumour and peritumoural immune infiltrates.

The single-channel tiff files for each selected image were extracted and the pixel intensities were rescaled from 0 to 1 with a custom R script. Using SIMPLI, single-cell segmentation was performed in each of the 35 images by applying a global threshold to the HOECHST channel to identify the nuclei and retain all objects with a diameter between 5 and 40 pixels. Each nucleus was then expanded by 5 pixels in all directions to define the cell area.

Resulting single cells were assigned to ten phenotypes according to the mean cell expression of CDX2 >0.15 or MUC1 >0.15 or cytokeratin >0.15 for tumour cells; CD34 >0.15 or CD31 >0.15 for endothelial cells; vimentin >0.1 for other stromal cells; CD11c >0.3 for dendritic cells; CD38 >0.26 for B cells; CD4 >0.13 and

CD3 >0.1 for CD4+ T cells; CD4 >0.12 and FOXP3 >0.5 and CD3 >0.1 for Tregs; CD8 >0.16 and CD3 >0.1 for CD8+ T cells, and CD68 >0.11 for macrophages. The heterotypic spatial analysis was performed by calculating the minimum distances of macrophages, CD8+ T cells, CD4+ T cells, Treg cells, and B cells to tumour cells and endothelial cells using the coordinates of the cell centroids. Only comparisons where the difference of the median cell–cell distances between the two histological subtypes was greater than 8 μm, corresponding to the diameter of B and T cells[30], were retained, no samples or cells were excluded from the analysis. As further support, a permutation test for each of the retained comparisons was run by re-assigning cell identities randomly in each sample 10,000 times. The resulting expected random distributions were compared to the observed values using a two-tailed permutation test and corrected for multiple testing.

**Reporting summary**. Further information on research design is available in the Nature Research Reporting Summary linked to this article.

## Data availability

The imaging mass cytometry data of human colon mucosa generated in this study have been deposited in the Zenodo database under accession code "5545882"[43]. The imaging mass cytometry data of the human appendix generated in this study have been deposited in the Zenodo database under accession code "5545760"[44]. The multiplex immunofluorescence data of human colorectal cancer generated in this study have been deposited in the Zenodo database under accession code "5545864"[45]. All other relevant data supporting the key findings of this study are available within the article and its Supplementary Information files or from the corresponding author upon reasonable request. Source Data are provided with this paper.

## Code availability

SIMPLI's code, documentation and an example dataset are available at "SIMPLI [https://github.com/ciccalab/SIMPLI]"[46]. The software code is protected by copyright. No permission is required from the rights-holder for non-commercial research uses. Commercial use will require a license from the rights-holder. For further information contact translation@crick.ac.uk who will reply within 5 business days.

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

## Acknowledgements

We thank Sharavan Vishaan Venkateswaran for testing SIMPLI. This work was supported by Cancer Research UK (C43634/A25487, F.D.C.), the Cancer Research UK King's Health Partners Centre at King's College London (C604/A25135, F.D.C.), the Cancer Research UK City of London Centre (C7893/A26233, F.D.C.), innovation programme under the Marie Skłodowska-Curie (grant agreement No CONTRA-766030, F.D.C.) and the Francis Crick Institute, which receives its core funding from Cancer Research UK (FC001002, F.D.C.), the UK Medical Research Council (FC001002, F.D.C.), and the Wellcome Trust (FC001002, F.D.C.) and Crohn's and Colitis UK (M2019/3, J.S.). For the purpose of Open Access, the authors have applied a CC BY public copyright licence to any Author Accepted Manuscript version arising from this submission.

## Author contributions

F.D.C. conceived and directed the study with support of J.S. M.B. developed the software with the help of D.T. and M.R.K. L.M. and A.A.-S performed the experiments. M.B, L.M, A.A.-S., M.J.P., J.S. and F.D.C. analysed the data. M.R.-J, G.B. and L.L. identified the samples and provided clinical assessments. M.R.-J. performed pathological assessments. M.B. and F.D.C. wrote the manuscript with contribution from A.A.-S., L.M. and M.J.P. All authors approved the manuscript.

## Competing interests

The authors declare no competing interests.
