## [Peer Review File · Nature Communications]

A SIMPLI (Single-cell Identification from MultiPLexed Images) approach for spatially-resolved tissue phenotyping at single-cell resolutionReviewers' Comments:

Reviewer #1:

Remarks to the Author:

Review of "A SIMPLI (Single-cell Identification from MultiPLexed Images) approach for spatially resolved tissue phenotyping at single-cell resolution" by M. Bortolomeazzi et al.

In their manuscript the authors have proposed SIMPLI as a multiplexed imaging analysis-software that performs single cell analysis for different types of multiplexed imaging modalities. To demonstrate the efficacy of their platform the authors have analyzed four different multiplexed data sets using SIMPLI.

Unfortunately, despite its potential, SIMPLI falls short of its high goals. Specifically,

1. Approaches based on cell and pixel level analysis have already been proposed (see, eg. <https://doi.org/10.1101/2021.01.05.425362>) and in my opinion, those proposed by the authors are not novel enough to merit publication.
2. Unifying image analysis pipelines have also been proposed (eg. <https://doi.org/10.1101/2021.03.15.435473>). Again, I am finding it difficult to see the fundamental improvement that SIMPLI provides.
3. Although the authors are proposing a generalized and technology-agnostic pipeline, their detailed description of SIMPLI shows that it is a collection of many available software tools/scripts (eg. EBImage, CellProfiler, Seurat, etc.) and requires many user- (or expert-) defined (ad-hoc) parameters like many of the current software technologies that the authors criticize.
4. The specific computational methods that have been proposed are not novel but have been known in image processing/analysis, computer vision literature, and do not provide any significant technology innovation. In fact, the CODEX paper (<http://www.ncbi.nlm.nih.gov/pmc/articles/pmc7479520/>), whose data the authors have utilized in their work, performs similar types of analyses. In my opinion the differences do not meet the novelty and significance threshold of Nature Communications.
5. I was quite surprised to see the short shrift given to cell segmentation. This is a major bottleneck that affects analyses of these images, and some seminal work has been done in this area (<https://doi.org/10.1101/2021.03.01.431313>). It is not clear how robust SIMPLI is in this regard.

Reviewer #2:

Remarks to the Author:

The authors describe a new containerised workflow for the analysis of spatial data derived from multiple platforms, including CODEX, IMC and mIF. With these technologies becoming more commonplace, this study is timely as it aims to streamline the downstream quantitative analyses. They showcase their tool using 4 case studies with diverse tissues and distinct biological questions, exemplifying the diversity of applications of SIMPLI. Overall this is a well-written and easy to read and understand manuscript.

Major comments:

- The main strengths of SIMPLI seem to be quick computing time and an automated pipeline that reduces manual input from users and processing time. However, it largely relies on other well-established tools that are used for each of the main steps in the workflow (such as CellProfiler, EBImage, Seurat), and the distances calculations between cells has been implemented in multiple other tools that the authors have highlighted in Table 1. The types of analyses suggested by SIMPLI have been featured in multiple other studies as well that the authors have largely referenced. This

work is likely be of most interest to computational biologists working with microscopy images or imaging facilities or labs. Significant major novel methodological contributions that expand our ability to understand spatial data were not clearly identifiable.

The authors discuss two modes of classification, using an unbiased approach with UMAPs and with expert input. Based on the manuscript, it was unclear why one was used over another, or how a binary threshold compared to that of an unbiased clustering approach. While the advantage of a clustering approach is not having to rely on expert input, in theory, it could also potentially help distinguish cells with different marker levels (e.g PDL1 high and PDL1 low expressing cells), that would be missed with a single threshold. However, the authors provide no metrics of the strengths of the unbiased clustering approach used by SIMPLI.

Normalisation of intensity is a generally a debated topic in the field. While normalisation has the advantage of being able to use similar thresholds across images or overlap clusters of cell phenotypes, the potential danger is that the level of background intensity and intensity of markers are often associated with tissue-specific properties (tissue age, length of time in fixation solution, among others). Therefore, a simple normalisation across tissues as used by the authors might over call or under call cell types. Additionally, often staining across a single tissue is not uniform, with better staining in some areas than others. The authors should provide a comparison of how the analysis of normalised data compares to that using raw data, tailoring thresholds for each image.

The pixel-based analysis is potentially a source of novelty, as it does not rely on cell segmentation, and provides information on extracellular or secreted proteins. The results they provide with secreted IgA are quite intriguing. What is the size of the IgA 'aggregates' that can be detected? It is unlikely to be single-protein resolution. While this is linked to the resolution at which the tissues were imaged, authors should touch on potentially artefacts and limitations of these analysis, including how SIMPLI distinguishes real secreted aggregates from dirt or other artefacts that lead to 'bright spots' in the images.

Selection of thresholds to identify cells positive for a marker in Figure 3F,G and Figure 4B,C are not convincing based on the distribution. The threshold seems to be arbitrary and roughly in the centre of the distribution, likely resulting in many false negative and false positive phenotypes. Could the authors expand on how they chose these thresholds, and how they have validated them?

The authors claim that there are significant differences in the distribution of distances from Tregs, B cells and macrophages to tumour cells and B cells to blood vessels with very significant p-values reported from Wilcoxon tests. However, visually, there are no convincing differences in the distributions. This is an often encountered problem in spatial analysis, where the large number of cells dramatically increases the power, resulting in seemingly significant differences. Did the authors calculate the distance for each cell in each of the 35 images, and then used that as their input to the Wilcoxon test, or did they calculate the average or median in each image and then performed the tests? The former is not ideal, given that each cell is not independent of each other and cannot be considered independent replicates. While in general I agree that the statistics for these types of analyses are still in their infancy and taking a non-parametric approach is a good proxy in most cases, I disagree with having this as a major result of the paper with such unconvincing differences in distributions.

Unfortunately due to technical reasons on my end unrelated to your software, I was unable to install and test it. Overall, based on the code repository, I commend the group for having a well-developed documentation. My general impression is that the usability of the software is highly unlikely to change my opinion regarding the content and value of the work for the community/readership of Nature Communications.

Minor comments:

The authors have selected relatively small ROIs for their analysis of 1 mm². However, most multiplex technologies allow staining and scanning of whole tissue sections. It is unclear how the authors selected specific ROIs in the images, and how their tool will scale up to larger ROIs covering entire tissue sections.

The the supplementary table 1, the legend is cutoff into two pages.

In Figure 4g, it is unclear what the highlighted cells correspond to.

Reviewer #3:

Remarks to the Author:

Bortolomeazzi et al present SIMPLI, an integrated multiplexed image analysis pipeline that can handle data from a range of different modalities. The authors first describe the workflow of their pipeline, which includes image preprocessing and normalization, single cell segmentation and phenotyping, and downstream spatial analysis. They then apply SIMPLI to a range of different imaging datasets, showing that their software is capable of analyzing data from many different sources. The authors have made SIMPLI publicly available via a well-documented github page.

Although the authors have created a one-stop-shop for all of the most common steps in analyzing imaging data, they have not adequately demonstrated that they solved the problem which has led to the current proliferation of different image analysis tools. In particular, the reason so many different approaches exist for analyzing imaging data is because of the peculiarities of imaging data from different sources. Given that no benchmarking was performed in this paper, it is not possible to evaluate how SIMPLI performs compared to these other, more limited analytical packages.

Major concerns:

1. The authors do not clearly delineate where they have developed novel techniques, and where they are using previously developed tools. For example, they use CellProfiler as their segmentation engine and Seurat for clustering. However, these are described in the same way as their spatial analysis code, which they have written themselves. This makes it challenging to identify which parts of the work represent novel contributions conceptually, and which are software engineering advancements meant to facilitate easy interoperability of existing tools.
2. Given that the field of image analysis is progressing so rapidly, the ability to swap out individual components of such a workflow is a crucial aspect. Segmentation is one example where the state of the art is advancing rapidly. The authors have chosen to use CellProfiler, instead of more accurate methods like StarDist, NucleAIzer, CellPose, or Mesmer. There are significant technical hurdles involved with integrating deep learning based segmentation into a pipeline; if the authors feel the costs outweigh the benefits, these should be discussed and justified in the text. Either way, a more detailed description of the modularity of their pipeline and examples for how to provide alternate containerized algorithms for key steps in the pipeline should be provided
3. The authors do not perform any benchmarking in the paper. Instead, to demonstrate that their approach works, they show a series of vignettes from different imaging modalities. As a methods paper, it is not necessary to discover novel biology in addition to demonstrating that their method performs well. However, in the absence of any benchmarking, the vignettes are the only source of confirmation that their pipeline produces reasonable results. As such, the fact that PD-L1 interacting T cells showed no change in cytotoxic markers (Figure 4e) does not represent the type of confident evidence needed to validate their approach. Similarly, the observation that Tregs are 6 um (approximately 1 cell length) closer to tumor cells in DII (Figure 5c) is not a sufficiently robust finding to validate their approach. In its current form, it is not possible to say whether their method actually works on imaging data from different modalities, or whether it simply produces results on imaging data from different modalities.

4. The authors appear to have systematically under-represented the ability of competing methods to work across image modalities in Table 1. As just two examples, both Ilastik and QuPath are capable of processing imaging data from all five modalities listed in the table. It is not clear how the authors populated this list, but it creates a highly biased impression of the relative advantage of their method.

Minor concerns

1. The authors should add Giotto

(<https://genomebiology.biomedcentral.com/articles/10.1186/s13059-021-02286-2>), and MCMICRO (<https://www.biorxiv.org/content/10.1101/2021.03.15.435473v1.full>) to Table 1

2. Image preprocessing is often platform specific, with different algorithms, approaches, and thresholds used for image data from different multiplexed imaging platforms. The authors have instead opted for a single preprocessing pipeline. The authors should validate that their selection of a single workflow for preprocessing is not inferior to the custom, platform-specific preprocessing employed for these different imaging techniques. In addition, some guidance for users about which parameters work well out of the box, and which will require fine-tuning, would be useful.

3. The authors make the following claim in a few places in the text: "Cells are assigned to populations using previously defined masks." However, it's not clear what this means. The description of cell classification in the methods section states that either unbiased clustering or user-defined thresholds are used to identify distinct populations. However, in Figure 2e it appears that cell populations have instead been defined using mask overlap. A more detailed description of what this step entails, and why it was chosen, is necessary to understand their approach

4. In general, the methods section is quite sparse, and does not describe the analytical workflow in sufficient detail to understand what the authors have done at each step. For example, the authors state that CellProfiler4 is used to generate thresholded images and masks, as well as the single-cell segmentation step, but do not go into further detail. It would be helpful to include more details here, such as which parameters need to be changed, how these masks are generated, and when one would use the different parts of the pipeline.

5. In the caption for Figure 2, the authors state that all cells overlapping with the lamina propria mask by at least 30% were considered resident. How was this threshold chosen and how does varying this threshold affect the results?

6. In Figure 2f, it would be more helpful to show stacked bar plots of the cell type breakdown (or somehow group the observations by CLN). This would allow the reader to better assess heterogeneity across CLNs.

7. In figure 3, the authors use both clustering methods to identify T cell subpopulations. However, they do not compare the results from these two approaches to show how closely they agree with one another.

8. In Figure 4g, the authors confirmed the localization of PDL1+CD68+ macrophages in close proximity to PD1+CD8+ cells by inspecting images, but the representative example is not clear. From where the arrow is pointing in the image inset, it looks as if the CD68 (pink) signal is adjacent to the PDL1 (green) signal, instead of being co-expressed in a single cell. Furthermore, the authors only phenotyped CD8+ T cells using their thresholding approach. To back-up the claim that PDL1+CD68+ macrophages are in close proximity to PD1+CD8+ cells, the authors could use unsupervised clustering or thresholding to quantify the number of CD68+ macrophages and quantify their distance to the PD1+CD8+ cells.

9. In the spatial analysis in Figure 5, the distance calculations of immune cells to tumor cells and blood vessels could be confounded by the number of tumor cells or the size of the epithelium in the ROI. It would be helpful here to perform a randomization of the cells in each ROI to generate a null distribution and assess whether the calculated distances are significant compared to a null distribution.

10. The authors state in the introduction that many competing approaches require "ad hoc configuration files." However, the authors use manually specified segmentation parameters in a configuration file as part of their analysis pipeline, and it appears that each dataset required a different set of parameters in order to produce accurate segmentations. This is likely the result of using Cell Profiler to perform segmentation, which is extremely sensitive to differences in image intensity and often requires manual tuning.

Reviewer #4:
None

Reviewer n.1

1. Approaches based on cell and pixel level analysis have already been proposed (see, eg. <https://doi.org/10.1101/2021.01.05.425362>) and in my opinion, those proposed by the authors are not novel enough to merit publication.

The tool mentioned by the Reviewer is described in a preprint, not yet peer reviewed, paper and the codes are not yet publicly available. This indicates that the field is fast moving and our method is as timely and novel as the others.

From the submitted manuscript we gather that the main differences with SIMPLI are:

- 1- At the cell level, this tool uses FlowSOM for clustering, which generally returns a very large number of clusters (100 in the cited preprint). These clusters then require extensive manual curation to derive biologically meaningful cell phenotypes. SIMPLI (which relies on Seurat for unsupervised clustering) allows the user to perform clustering at different levels of resolution in a single run and returns visualisations plots to help identify the most biologically meaningful phenotypes.
- 2- At the pixel level, the cited method applies unsupervised clustering to pixel values. SIMPLI instead measures pixel positive areas for all or user-defined combinations of markers. Additionally, these positive areas can be normalised over the area of the whole image, specific tissue compartments or cell populations for direct comparisons across samples and conditions.

For these reasons we think that the tools are sufficiently different. Moreover, other existing approaches enabling analyses at cell and pixel level all have limits in terms of technologies and analytical steps that they can cover. These methods are all listed in Table 1 and their limits are further explained in the Introduction (p. 3-4).

2. Unifying image analysis pipelines have also been proposed (eg. <https://doi.org/10.1101/2021.03.15.435473>).

The approach cited by the reviewer (MCMICRO) is again described in a preprint, not yet peer-reviewed paper. Also in this case, it is significantly different from SIMPLI in terms of:

- 1- Tools implemented in the two pipelines, which have no overlap.
- 2- Possibility offered by SIMPLI (but not MCMICRO) to skip specific analytical steps and be use alternative input at each step of the analysis. This is now explained in the Results (p. 7), new Supplementary Fig.1 and software documentation in Github.
- 3- Requirements by MCMICRO of a specific file directory structure, thus limiting flexibility. SIMPLI instead only requires metadata files than can be edited with any text editor. This is now explained in the Introduction (p. 4), in the Methods (p. 31), and in the software documentation in Github.

3. Although the authors are proposing a generalized and technology-agnostic pipeline, their detailed description of SIMPLI shows that it is a collection of many available software tools/scripts (eg. EBImage, CellProfiler, Seurat, etc.) and requires many user- (or expert-) defined (ad-hoc) parameters like many of the current software technologies that the authors criticize.

SIMPLI integrates novel and well-established tools, which all require custom codes to be run as standalone software. This is a limit for users with limited computational background. SIMPLI overcomes this limit by automatically formatting input and output of each tool to make them interoperable, without the need of user intervention but still allowing the user to choose specific parameters if desired. This is now explained in the Introduction (p. 4), Results (p. 7), Methods (p. 31-33) and Supplementary Fig.1)

It should be noted that, of all the tools used by SIMPLI, CellProfiler is the only one requiring user-defined parameters. This step cannot be avoided because it depends on the specifics of the experiment (tissue type, imaging technology, etc.). However, parameters can be configured through CellProfiler's graphical interface and the resulting metadata can then be easily imported into SIMPLI. Similarly, the expert-guided definition of thresholds cannot be avoided because it is related to the nature of the tissues under investigation.

4. The specific computational methods that have been proposed are not novel but have been known in image processing/analysis, computer vision literature, and do not provide any significant technology innovation. In fact, the CODEX paper (<http://www.ncbi.nlm.nih.gov/pmc/articles/pmc7479520/>), whose data the authors have utilized in their work, performs similar types of analyses. In my opinion the differences do not meet the novelty and significance threshold of Nature Communications.

As explained in previous points, SIMPLI introduces novel tools and integrates them with well-established ones (now clarified in Supplementary Fig.1 and in the text).

In the specific case of the CODEX Toolkit, it is a pipeline for the cell level analysis of CODEX images only. We show that SIMPLI can not only perform similar analysis (Figure 5a,b) but also add the cell spatial characterisation (Figure 5 c-f) that the CODEX Toolkit does not provide.

5. I was quite surprised to see the short shrift given to cell segmentation. This is a major bottleneck that affects analyses of these images, and some seminal work has been done in this area (<https://doi.org/10.1101/2021.03.01.431313>). It is not clear how robust SIMPLI is in this regard.

The method cited by the reviewer is again described in a pre-print manuscript and we do think it is appropriate to implement it into SIMPLI before it has been fully peer reviewed.

To address the point of the Reviewer (see also response to point 2 of Rev. 3), in addition to CellProfiler, we have now implemented the option to perform segmentation with StarDist, which is based on machine learning and uses star-convex polygons to represent cell shapes¹. It is therefore a completely different approach than CellProfiler.

We have expanded the description of the cell segmentation step in SIMPLI (p. 7) and added a comparison between StarDist and CellProfiler (Supplementary Figs. 1, 3a; Results p. 16; Methods p. 31).

Reviewer n.2

Overall this is a well-written and easy to read and understand manuscript.

We thank the reviewer and are glad that they found the manuscript clear.

Major comments:

1- The authors discuss two modes of classification, using an unbiased approach with UMAPs and with expert input. Based on the manuscript, it was unclear why one was used over another, or how a binary threshold compared to that of an unbiased clustering approach. While the advantage of a clustering approach is not having to rely on expert input, in theory, it could also potentially help distinguish cells with different marker levels (e.g PDL1 high and PDL1 low expressing cells), that would be missed with a single threshold. However, the authors provide no metrics of the strengths of the unbiased clustering approach used by SIMPLI.

The two modes of cell classification (phenotyping) implemented in SIMPLI are not mutually exclusive and the user can prefer one over the other depending on their scientific questions and experimental setting. The two approaches can also be used to cross-validate each other, as shown for the analysis of PD1⁺ T follicular helper cells (Fig. 3d-h). We clarify this in the text (p.7), Fig. 1 and Supplementary Fig. 1.

To compare cell classifications obtained with binary thresholding and unbiased clustering, we extended the analysis of all T cells unsupervised clusters shown in Fig. 3d,e and confirmed that they are largely replicated using expert-defined thresholds (Supplementary Fig. 3b,c,d). This is now commented in the text (p. 17).

To provide further evidence of the robustness of the unbiased clustering, we compared clusters at different levels of resolutions (0.25, 0.50, 1). All three runs of clustering group cells into similar populations. However, the five clusters shown in Fig. 3e (0.25 resolution) were split into smaller subgroups when using higher resolutions. For example, as suggested by the reviewer, CD4⁺PD1⁺ T cells composing cluster 5 in Fig.3d-e were further split in CD4⁺ PD1 high and PD1 low cells when using a resolution of 1 (Supplementary Fig. 3a). This is now extensively commented in the Results (p. 16-17), Methods (p. 37), revised Fig.3 and Supplementary Fig.3a. Finally, the possibility of choosing different levels of resolution for unsupervised clustering classification is now fully explained in the software documentation in Github

2- Normalisation of intensity is a generally a debated topic in the field. While normalisation has the advantage of being able to use similar thresholds across images or overlap clusters of cell phenotypes, the potential danger is that the level of background intensity and intensity of markers are often associated with tissue-specific properties (tissue age, length of time in fixation solution, among others). Therefore, a simple normalisation across tissues as used by the authors might over call or under call cell types. Additionally, often staining across a single tissue is not uniform, with better staining in some areas than others. The authors should provide a comparison of how the analysis of normalised data compares to that using raw data, tailoring thresholds for each image.

We shall note that in SIMPLI normalisation is performed separately for each marker in each sample by rescaling all pixel values of each channel up to the 99th percentile of

the distribution. It therefore should not be affected by tissue properties. Normalised images are then used to identify unique thresholds that are applied across samples. In addition, this step can be skipped by the user if desired. We clarified this in the Results (p. 7), revised Fig.1a and software documentation in GitHub.

To address the point raised by the reviewer, we compared the pixel-level IgA distribution in normal colon mucosa with and without normalisation. This comparison showed a strong linear correlation between values of IgA⁺ areas from normalised and raw data (Supplementary Fig. 2c). This is now described in the Results (p. 12) and Methods (p. 36).

3- The pixel-based analysis is potentially a source of novelty, as it does not rely on cell segmentation, and provides information on extracellular or secreted proteins. The results they provide with secreted IgA are quite intriguing. What is the size of the IgA 'aggregates' that can be detected? It is unlikely to be single-protein resolution. While this is linked to the resolution at which the tissues were imaged, authors should touch on potentially artefacts and limitations of these analysis, including how SIMPLI distinguishes real secreted aggregates from dirt or other artefacts that lead to 'bright spots' in the images.

The resolution of imaging mass cytometry (1 μ m²) unfortunately does not allow single-molecule detection. It is however compatible with detection of secreted IgA aggregates, as shown in Figure 2b. We have several indications that these are real aggregates and not artefacts:

- 1- Secreted IgA are usually formed of IgA dimers bound to a joining protein (J chain) and a secretory component which facilitates transcytosis through the mucosal epithelium. This process occurs preferentially in the epithelial crypts². We now added histological evidence to show that secreted IgAs indeed localise in the epithelial crypts with only a minimal contribution of IgA⁺ area in the surface epithelium (Supplementary Figure 2a, Results p. 11-12).
- 2- Bright pixel artefacts are removed during the image pre-processing step by applying Gaussian smoothing with a radius of 1.5 pixel before thresholding the IgA signal. This is now further explained in the Methods (p. 35-36).
- 3- Remaining artefact located outside the tissue masks (an example is provided in Supplementary Figure 2a) are not retained because they do not overlap with any compartment mask (Supplementary Figure 2b).

4- Selection of thresholds to identify cells positive for a marker in Figure 3F,G and Figure 4B,C are not convincing based on the distribution. The threshold seems to be arbitrary and roughly in the centre of the distribution, likely resulting in many false negative and false positive phenotypes. Could the authors expand on how they chose these thresholds, and how they have validated them?

We agree with the Reviewer that the distribution of the marker expression across cells is not always reliable, particularly when it is not bimodal and thus does not indicate a clear value for the threshold. This is why thresholds for each marker were selected after expert visual inspection of the original images. We have expanded the general explanation of this approach in the manuscript (p. 8) and specifically for Figure 3f,g (p. 17 and 37).

We now provide extensive support showing that cells identified by expert-defined thresholds and those identified by unsupervised clustering share very similar expression profiles (Figure 3e-h, Supplementary Fig.3b-d).

Also in the case of Figure 4b,c we applied user-defined thresholds. In the new Figure 4i (see response to minor point 3) we provide example of cells whose identity has been assigned with this approach and that indeed express all relevant markers.

5- The authors claim that there are significant differences in the distribution of distances from Tregs, B cells and macrophages to tumour cells and B cells to blood vessels with very significant p-values reported from Wilcoxon tests. However, visually, there are no convincing differences in the distributions. This is an often encountered problem in spatial analysis, where the large number of cells dramatically increases the power, resulting in seemingly significant differences. Did the authors calculate the distance for each cell in each of the 35 images, and then used that as their input to the Wilcoxon test, or did they calculate the average or median in each image and then performed the tests? The former is not ideal, given that each cell is not independent of each other and cannot be considered independent replicates. While in general I agree that the statistics for these types of analyses are still in their infancy and taking a non-parametric approach is a good proxy in most cases, I disagree with having this as a major result of the paper with such unconvincing differences in distributions.

The comparison of distances between cell populations in the two CRC subgroups was done in two rounds:

First, the distributions of distances between five types of immune cells (macrophages, Tregs, CD8+, CD4+, and B cells) and blood vessels or tumour cells were compared with a non-parametric two-tailed Wilcoxon test, for a total of ten comparisons. After FDR correction, six comparisons resulted significant (Tregs to tumour cells; CD8 T cells to blood vessels; macrophages to tumour cells and blood vessels; B cells to tumour cells and blood vessels).

Second, to take into account the fact that the large number of cells dramatically increases the test statistical power (*i.e.* the concern raised by the reviewer), we retained only significant comparisons where the median cell-cell distances was greater than 4 μ m. This roughly corresponds to the radius of B and T lymphocytes³ and reduced the number of significant comparisons to the four previously shown in Figure 5c,d (Tregs to tumour cells; macrophages to tumour cells; B cells to tumour cells and blood vessels).

We now decided to be even more restrictive and increased the difference of median cell-cell distances between the CRC subtypes to 8 μ m, corresponding to the diameter of B and T cells. We reasoned that this would mean difference of at least a cell monolayer. With this new cut-off only two of the comparisons are retained (Tregs to tumour cells and B cells to blood vessels, new Figure 5c,d).

Moreover, we added a permutation test (see also response to point 3 and minor point 9 of Rev. 3) where we compared the observed difference to the distribution of expected differences. This further analysis confirmed the significance of these observations (Figure 5e,f)

We now provide more detail on this in the Results (p. 25-26), Methods (p. 40) and modified Figure 5c-f.

Minor comments:

1- The authors have selected relatively small ROIs for their analysis of 1 mm². However, most multiplex technologies allow staining and scanning of whole tissue sections. It is unclear how the authors selected specific ROIs in the images, and how their tool will scale up to larger ROIs covering entire tissue sections.

For Figures 2 and 3, the ROIs were indeed of 1mm², selected as follows:

- For the six colon mucosa samples (Figure 2), ROIs were selected to include the whole structure of the mucosa with the epithelial crypts in longitudinal orientation.
- For the appendix (Figure 3), the ROI was selected to contain a whole lymphoid follicle along with the surrounding epithelium and lamina propria. The selection criteria are now described in the Methods (p. 35).

For Figure 4, the ROI was >5mm² and we report the performance of SIMPLI in Table 2. We now also comment on this in the Results (p. 20), Methods (p. 39), and in the Discussion (p. 30).

For Figure 5, we use ROIs of approximately 1mm², as in the original publication.

2- The the supplementary table 1, the legend is cutoff into two pages.

This is because of the automatic conversion into pdf files. The supplementary tables are also available as original xls files where all legends are properly formatted.

3- In Figure 4g, it is unclear what the highlighted cells correspond to.

Figure 4g showed an example of CD8⁺PD1⁺ T cells in proximity to PDL1⁺CD68⁺ macrophages.

We agree with the reviewer that this was not the clearest example and have now replaced the panel (now Figure 4i) to improve clarity. In the new panel we show key markers first individually and then in a merged image. We also added a better description in the figure legend.

Reviewer n.3

Given that no benchmarking was performed in this paper, it is not possible to evaluate how SIMPLI performs compared to these other, more limited analytical packages.

At present, a comprehensive benchmark of SIMPLI is not possible because the other potentially comparable software that also combine all analytical steps are not yet available. Moreover, the individual tools included in SIMPLI (CellProfiler, EBIImage, Seurat) are all widely adopted for the analysis of imaging or single cell data.

However, we added further supports to the robustness of SIMPLI in the response to the individual points below.

Major concerns:

1. The authors do not clearly delineate where they have developed novel techniques, and where they are using previously developed tools. For example, they use CellProfiler as their segmentation engine and Seurat for clustering. However, these are described in the same way as their spatial analysis code, which they have written themselves. This makes it challenging to identify which parts of the work represent novel contributions conceptually, and which are software engineering advancements meant to facilitate easy interoperability of existing tools.

SIMPLI is indeed a combination of well-established and newly developed tools. We have now clarified this in the Results (p. 7) and Methods (p 31-33) and have highlighted custom codes vs established tools in in Supplementary Fig.1 and associated legend.

2. Given that the field of image analysis is progressing so rapidly, the ability to swap out individual components of such a workflow is a crucial aspect. Segmentation is one example where the state of the art is advancing rapidly. The authors have chosen to use CellProfiler, instead of more accurate methods like StarDist, NucleAIzer, CellPose, or Mesmer. There are significant technical hurdles involved with integrating deep learning based segmentation into a pipeline; if the authors feel the costs outweigh the benefits, these should be discussed and justified in the text. Either way, a more detailed description of the modularity of their pipeline and examples for how to provide alternate containerized algorithms for key steps in the pipeline should be provided

Following the Reviewer suggestion, in addition to CellProfiler, we have now implemented the option to perform segmentation also with StarDist, which is based on machine learning and is therefore a completely different approach than CellProfiler.

We selected StarDist over the others cited by the reviewer because it provides three pre-trained models that the user can test directly on their data thus avoiding the labour-intensive requirement to train a model specific to their analysis. However, SIMPLI provides the option to run StarDist with user generated models on a custom set of markers, and thus apply this method to its full potential. Additionally, SIMPLI allows the selection of custom values of the probability and non-maximum suppression thresholds for both the pre-trained models bundled with StarDist and user generated ones. This is fully explained in the software documentation in GitHub

We have expanded the description of the cell segmentation step in SIMPLI (p. 7) and added a comparison between StarDist and CellProfiler (Supplementary Fig. 3a p. 7, 16, and Methods p. 32).

3. The authors do not perform any benchmarking in the paper. Instead, to demonstrate that their approach works, they show a series of vignettes from different imaging modalities. As a methods paper, it is not necessary to discover novel biology in addition to demonstrating that their method performs well. However, in the absence of any benchmarking, the vignettes are the only source of confirmation that their pipeline produces reasonable results. As such, the fact that PD-L1 interacting T cells showed no change in cytotoxic markers (Figure 4e) does not represent the type of confident evidence needed to validate their approach.

We explain in response to point 1 why benchmarking is not possible.

To the best of our knowledge, the spectrum of cytotoxic activity in exhausted CD8⁺PD1⁺ T cells that has been described in the literature⁴ is in line with what we show in Figure 4e.

Additionally, this is supported by the direct observation that CD8⁺PD1⁺ T cells expressing granzyme B can be equally found in proximity to or distant from PDL1⁺ cells (Figure 4i). We have now commented on this in the Results (p. 21).

Similarly, the observation that Tregs are 6 μm (approximately 1 cell length) closer to tumor cells in DII (Figure 5c) is not a sufficiently robust finding to validate their approach. In its current form, it is not possible to say whether their method actually works on imaging data from different modalities, or whether it simply produces results on imaging data from different modalities.

The difference in distances between Tregs and tumour cells in DII and CLR subtypes in Figure 5c is $>13\mu\text{m}$, which is twice the cell length.

To further support this observation, we now run a permutation test (see also response to minor comment 9) where we compared the observed difference to the distribution of expected differences. This further analysis confirmed the significance of the observation (Figure 5e)

4. The authors appear to have systematically under-represented the ability of competing methods to work across image modalities in Table 1. As just two examples, both Ilastik and QuPath are capable of processing imaging data from all five modalities listed in the table. It is not clear how the authors populated this list, but it creates a highly biased impression of the relative advantage of their method.

In the previous version of the manuscript, for each method we restricted the image technologies to those explicitly reported in the original publication.

To address the reviewer's comment, we performed a literature search for each method in Table 1 and included all imaging technologies for which there was published evidence of images analysed with that method (modified Table 1). We now describe these criteria in the legend to Table 1 (p. 6).

Minor concerns

1. The authors should add Giotto (<https://genomebiology.biomedcentral.com/articles/10.1186/s13059-021-02286-2>), and MCMICRO (<https://www.biorxiv.org/content/10.1101/2021.03.15.435473v1.full>) to Table 1

We have added GIOTTO to Table 1, but not MCMICRO because it is a preprint, not yet peer reviewed method. Since GIOTTO works predominantly with spatial transcriptomic data, we have added this technology to the list of Table 1.

2. Image preprocessing is often platform specific, with different algorithms, approaches, and thresholds used for image data from different multiplexed imaging platforms. The authors have instead opted for a single preprocessing pipeline. The authors should validate that their selection of a single workflow for preprocessing is not inferior to the custom, platform-specific preprocessing employed for these different imaging techniques. In addition, some guidance for users about which parameters work well out of the box, and which will require fine-tuning, would be useful.

Imaging pre-processing in SIMPLI is composed of three processes (Fig.1a and Supplementary Fig.1a):

1. data extraction from .mdc or .txt files to obtain raw images (limited to MIBI and IMC)
2. data normalisation (optional)
3. thresholding and masking with CellProfiler4⁵. This involves either identifying unique thresholds to reduce background noise across samples (if step 2 has been retained) or deriving sample-specific thresholds to minimise the effect of non-uniform staining in individual raw images (if step 2 has been skipped). The selection of the thresholds is experiment-specific.

This approach to data pre-processing makes SIMPLI highly flexible and thus suitable for the analysis of images produced with different multiplexed-imaging technologies. For instance, we were able to pre-process CODEX derived images obtaining similar results of the original publication (Figure 5a,b).

We have now expanded our explanation of the pre-processing step in the main text (p. 7) and in the methods (p. 31).

As requested by the Reviewer, we also provide a description of the default parameters, the documentation on the CellProfiler4 pipeline, and an explanation of the analysis run on the example dataset in the software documentation in GitHub.

3. The authors make the following claim in a few places in the text: “Cells are assigned to populations using previously defined masks.” However, it’s not clear what this means. The description of cell classification in the methods section states that either unbiased clustering or user-defined thresholds are used to identify distinct populations. However, in Figure 2e it appears that cell populations have instead been defined using mask overlap. A more detailed description of what this step entails, and why it was chosen, is necessary to understand their approach

We acknowledge we this point was not explained properly.

Cell classification (which we refer to as *phenotyping* throughout the text) is indeed done using either unsupervised clustering or applying user-defined thresholds.

However, the starting cells can be either all single cells obtained after cell segmentation or only those mapping to tissue compartments (i.e. epithelium) or positive for certain markers (i.e. CD3). In other words, cell phenotyping can be done starting from all T cells or epithelial cells rather than from all cells in the image.

Assignment to a specific tissue compartment or positivity to a certain marker derive from the overlap of the single cell masks with the tissue compartment or marker masks derived at the end of the pre-processing step.

We have now explained this in the Results (p. 8), in revised Supplementary Figure 1b, in its legend and throughout the text, where we replaced the reference to 'assignment to cell populations' with 'positivity to markers'.

4. In general, the methods section is quite sparse, and does not describe the analytical workflow in sufficient detail to understand what the authors have done at each step. For example, the authors state that CellProfiler4 is used to generate thresholded images and masks, as well as the single-cell segmentation step, but do not go into further detail. It would be helpful to include more details here, such as which parameters need to be changed, how these masks are generated, and when one would use the different parts of the pipeline.

We have now expanded the Methods significantly to provide more details on:

- the general analytical workflow (p. 31-33), Supplementary Figure 1 and legend
- how each of the four exemplar analyses was conducted (p. 35-40).

Additionally, we added details on parameter selection in the software documentation in GitHub

5. In the caption for Figure 2, the authors state that all cells overlapping with the lamina propria mask by at least 30% were considered resident. How was this threshold chosen and how does varying this threshold affect the results?

As explained in response to point 4 of Rev. 2 and in the text (p. 8), all thresholds, including this one, were chosen by an expert histopathologist (JS) after reviewing the raw images together with the cell segmentation masks and the tissue compartment masks (in this case the mask of the lamina propria).

To assess how this threshold may have affected the results, we have now repeated the same analyses shown in Figure 2e,f after assigning cells to the lamina propria at various values of thresholds.

This analysis showed that the only cells changing assignment were those at the boundary between lamina propria and epithelium (Supplementary Figure 2d). They constituted around 10% of all cells and therefore varying these thresholds had a limited impact on the overall proportion of lamina cells (Supplementary Figure 2e) as well as of resident immune cells (Supplementary figure 2f). We now report these observations in the Results (p. 12) and expanded on the selection of this threshold in the Methods (p. 37).

6. In Figure 2f, it would be more helpful to show stacked bar plots of the cell type breakdown (or somehow group the observations by CLN). This would allow the reader to better assess heterogeneity across CLNs.

We have now replaced Figure 2f and associated legend with a stacked bar plot of immune cell proportion as suggested.

7. In figure 3, the authors use both clustering methods to identify T cell subpopulations. However, they do not compare the results from these two approaches to show how closely they agree with one another.

See also response to Point 1 of Rev. 2. We have now compared systematically the results of both phenotyping approaches in classifying CD4⁺PD1⁺ T cell subpopulations, showing that they are overall comparable (Supplementary figure 3b-d). We comment extensively on this in the text (p. 18) and in the Methods (p. 36).

8. In Figure 4g, the authors confirmed the localization of PDL1⁺CD68⁺ macrophages in close proximity to PD1⁺CD8⁺ cells by inspecting images, but the representative example is not clear. From where the arrow is pointing in the image inset, it looks as if the CD68 (pink) signal is adjacent to the PDL1 (green) signal, instead of being co-expressed in a single cell. Furthermore, the authors only phenotyped CD8⁺ T cells using their thresholding approach. To back-up the claim that PDL1⁺CD68⁺ macrophages are in close proximity to PD1⁺CD8⁺ cells, the authors could use unsupervised clustering or thresholding to quantify the number of CD68⁺ macrophages and quantify their distance to the PD1⁺CD8⁺ cells.

See also response to Point 4 of Rev. 2. We have now replaced the old panel with clearer example of CD8⁺PD1⁺ T cells in proximity to a PDL1⁺CD68⁺ macrophage (Figure 4i).

To address the second point raised by the Reviewer, we have identified PDL1⁺ and PDL1⁻ CD68⁺ macrophages by expression thresholding (Figure 4g). We then measured their respective distances to CD8⁺PD1⁺ T cells confirming that PDL1⁺ macrophages are significantly closer to CD8⁺PD1⁺ T cells than PDL1⁻ macrophages (Figure 4h). We describe this additional validation in the main text (p. 20-21) and the methods (p. 39) and new Figure 4.

9. In the spatial analysis in Figure 5, the distance calculations of immune cells to tumor cells and blood vessels could be confounded by the number of tumor cells or the size of the epithelium in the ROI. It would be helpful here to perform a randomization of the cells in each ROI to generate a null distribution and assess whether the calculated distances are significant compared to a null distribution.

As suggested by the Reviewer, we now performed a permutation test for each of the ten comparisons by re-assigning the cell identities randomly in each sample 10.000 times. At each permutation, we calculated the median distances of the five immune populations (macrophages, CD8⁺ T cells, CD4⁺ T cells, Tregs and B cells) to tumour cells or vessels, between CLR and DII samples. We derived ten expected distributions, compared each of them to the observed values using a two-tailed permutation test and corrected for multiple testing.

We confirmed that T-regs are closer to tumour cells in the DII subtype (Figure 5e), while B cells are closer to vessels in the DII subtype (Figure 5f), as compared to the corresponding expected distributions. We comment this in the Results (p. 26)

10. The authors state in the introduction that many competing approaches require “ad hoc configuration files.” However, the authors use manually specified segmentation parameters in a configuration file as part of their analysis pipeline, and it appears that each dataset required a different set of parameters in order to produce accurate segmentations. This is likely the result of using Cell Profiler to perform segmentation, which is extremely sensitive to differences in image intensity and often requires manual tuning.

See also response to Point 3 of Rev. 1.

Of all the tools used by SIMPLI, CellProfiler is the only one requiring user-defined parameters. This step cannot be avoided because it depends on the specifics of the experiment (tissue type, imaging technology, etc.). However, parameters can be configured through CellProfiler’s graphical interface and the resulting metadata can then be easily imported into SIMPLI.

Now we have added the option to perform cell-segmentation with StarDist. This removes the need for the user to tune parameters if using one of the default pretrained models, while still giving the possibility to use *ad hoc* models for use cases with highly specific requirements.

References

1. Schmidt, U., Weigert, M., Broaddus, C. & Myers, G. Cell Detection with Star-Convex Polygons. 265-273 (Springer International Publishing, Cham, 2018).
2. Bjerke, K. & Brandtzaeg, P. Lack of relation between expression of HLA-DR and secretory component (SC) in follicle-associated epithelium of human Peyer's patches. *Clinical and experimental immunology* **71**, 502-507 (1988).
3. Strokotov, D. *et al.* Is there a difference between T- and B-lymphocyte morphology? *Journal of Biomedical Optics* **14**, 064036 (2009).
4. Zhang, L. *et al.* Lineage tracking reveals dynamic relationships of T cells in colorectal cancer. *Nature* **564**, 268-272 (2018).
5. McQuin, C. *et al.* CellProfiler 3.0: Next-generation image processing for biology. *PLOS Biology* **16**, e2005970 (2018).

Reviewers' Comments:

Reviewer #1:

Remarks to the Author:

The authors have satisfactorily answered by questions and concerns. I particularly agree with their argument regarding comparing their work with preprints, which makes their work timely. Some of my other concerns were answered by them in their response to other reviewers. I, therefore, recommend their manuscript for publication.

Reviewer #2:

Remarks to the Author:

While the authors generally address the technical aspects brought up under review, they conveniently skip the points brought up by the reviewers regarding the lack of novelty of the work. This work is mainly a collection of already well-established tools and does not advance the field of spatial analysis. Furthermore, the use of each of the tools described is actually easier than the workflow proposed by SIMPLI, which makes it difficult to think that this new tool will be widely adopted.

Technical issues:

For point number 2 of reviewer 2 comments:

If normalization is done within image alone, then this does not correspond to an adequate normalization, as the normalized values would be highly dependent on the cells present. For example, if there are no cells of a particular type, but there is some background non-specific staining for this marker, the rescaling done by SIMPLI will lead to false positives. It is also surprising that the authors claim in their paper "that data normalisation has no impact on the results" – which begs the question of why include this normalization at all if there are no practical benefits.

For point 5 of reviewer 2 comments:

The updates provided do not address the issues raised. As explained previously, it is not statistically correct to combine cells across different tissue sections and then perform statistical tests (regardless of the filtering the authors have done). This is because cells in an image are not independent of each other. For example, if you have 2 images, one with 500 cells, and the other with 400 cells, following the procedure set out by the authors, there would be '900 samples' (minus some lost by filtering), but in reality, there are only 2 samples. Therefore, their procedure is inadequate and artificially inflates the sample size. Additionally, the use of permutation tests does not support their claims, as permutation tests just test against complete spatial randomness, and biological tissues are not 'random' – therefore, you will likely always get significant p-values.

Minor issues:

Table I contains many inaccuracies. For example, SPIAT does in fact allow unsupervised cell phenotyping as outlined in their manuscript.

Reviewer #3:

Remarks to the Author:

Bortolomeazzi et al

We thank the authors for responding in detail to the points raised in our initial round. We would like to emphasize that the point of addressing the comments below is not to demonstrate the pipeline the authors have developed is perfect in every way, but to better characterize its strengths and weaknesses. We feel it is important to understand what the boundary conditions using SIMPLI are given specific data types and programmatic assumptions that are being made. Since a big part of the

value is to unify and simplify image analysis, the end users that will most significantly benefit are groups with less experience in dealing with these datasets. Consequently, it is important for the authors to articulate where SIMPLI should perform well and in what scenarios it is more prone to erroneous results. With this in mind, we have listed below the areas where we have remaining concerns:

- The authors stated that they do not want to add MCMICRO to Table 1 because it is not yet peer reviewed. We agree that the fact a study has not yet been published is a reasonable counterpoint to the novelty concerns raised by reviewer 1. However, given that preprints are citeable, we do not feel that it justifies exclusion from surveys of available tools, as many methods (especially computational ones) are available via preprint, and widely used, well in advance of final publication.
- We share the concerns of reviewer 2 about the normalization technique the authors are proposing. Tissue-specific and platform-specific differences can influence which normalization techniques are used. The fact that the authors have performed channel-specific normalization does not mean that these issues have disappeared. For example, tissues or channels with particularly low signal-to-noise ratios may not work well with the same percentile cutoff used for the rest of a dataset. Furthermore, this cutoff may be different across different image platforms.
- This is related to our point about benchmarking, as the majority of our concerns centered around preprocessing. We are not asking the authors to evaluate every aspect of their pipeline, since as they correctly note, it encompasses many previously validated individual tools. However, given that different imaging platforms often perform different normalization schemes, the authors' claims that their approach can be used to uniformly process data from all of them merits a more detailed evaluation, as this is a departure from the way some of these data are usually analyzed. In particular, the authors should demonstrate across all of the imaging platforms included that the use of their normalization scheme yields equivalent results to the normalizations commonly used for those data types. This is particularly important since there is no data presented from three of the technologies the authors say their method works on (mIHC, MIBI, and spatial transcriptomics).
- The permutation test analysis provides convincing support for the results of the spatial analysis, but it would be helpful to include it in the SIMPLI pipeline to allow users to assess the significance of their results.
- The authors apply a Gaussian filter to the IgA images for pixel analysis. It is not clear why this threshold was chosen and if this is a tunable parameter in SIMPLI.

Reviewer n. 1

The authors have satisfactorily answered by questions and concerns. I particularly agree with their argument regarding comparing their work with preprints, which makes their work timely. Some of my other concerns were answered by them in their response to other reviewers. I, therefore, recommend their manuscript for publication.

We thank the reviewer for their time and useful feedback on our work.

Reviewer n. 2

While the authors generally address the technical aspects brought up under review, they conveniently skip the points brought up by the reviewers regarding the lack of novelty of the work. This work is mainly a collection of already well-established tools and does not advance the field of spatial analysis. Furthermore, the use of each of the tools described is actually easier than the workflow proposed by SIMPLI, which makes it difficult to think that this new tool will be widely adopted.

As stated in the manuscript (p. 6), SIMPLI integrates a collection of well-established tools and newly developed functionalities, such as the pixel analysis and the spatial analysis, as acknowledged by this Reviewer in the first round of revisions (point 3: *“The pixel-based analysis is potentially a source of novelty, as it does not rely on cell segmentation, and provides information on extracellular or secreted proteins”*).

Moreover, as pointed out by Reviewer n. 3, SIMPLI facilitates the compatibility across tools making them more accessible for non-computational users. This is described in the Introduction (p. 4)

Technical issues:

For point number 2 of reviewer 2 comments:

1. If normalization is done within image alone, then this does not correspond to an adequate normalization, as the normalized values would be highly dependent on the cells present. For example, if there are no cells of a particular type, but there is some background non-specific staining for this marker, the rescaling done by SIMPLI will lead to false positives.

In point 2 of the previous round of comments, the Reviewer asked us to *“provide a comparison of how the analysis of normalised data compares to that using raw data, tailoring thresholds for each image”*. This is indeed what we did, showing that IgA⁺ areas from normalised and raw data correlate positively (Supplementary Fig. 2c).

In the new case described by the Reviewer now (i. e. *“if there are no cells of a particular type, but there is some background non-specific staining for this marker”*), it should be noted that normalisation in SIMPLI is followed by a cleaning step where the user can:

- 1- Apply multiple filters to remove high intensity sparkles (see point 5 of R. 3).
- 2- Specify a threshold to each marker after expert review of the images.

This is described in the Results (p. 6), Methods (p. 31) and in the GitHub documentation. Moreover, as explained in the text (p. 6), the normalisation step is optional and can be skipped (on its benefits, see next point).

To further test the effects on normalisation and in particular whether it leads to false positives, we compared the results of each of the four case studies reported in the paper applying SIMPLI's normalisation with those applying alternative approaches:

- IgA⁺ pixel analysis from IMC data: We have compared IgA distribution with and without normalisation showing a strong linear correlation (Supplementary Fig. 2c). This suggests that normalisation does not affect the results. This has been described in the Results (p. 11) and Methods (p. 36).
- Relative proportion of cell types from IMC data: We show that the proportions of T cells, B cells, macrophages, dendritic cells and epithelial cells as obtained after SIMPLI's normalisation (Fig. 3c) are comparable to those obtained by applying HistoCAT Z-score normalisation (Supplementary Fig. 3b). This is now commented in the Results (p. 15).
- Relative proportions of cell types from mIF data: we obtained similar proportion of PD1⁺CD8⁺ cells and PDL1⁺ cells after applying SIMPLI normalisation (Fig. 4d) or the Inform tissue analysis software¹ distributed with the Vectra Polaris automated quantitative pathology imaging system (Akoya Biosciences) (Fig. 4e). This is now commented in the Results (p. 20).
- Relative proportion of cell populations from CODEX data: We started from the raw CODEX data and, after data normalisation, we obtained similar proportions of immune cells as in the original paper (Fig. 5b). This is discussed in the Results (p. 25) and Methods (p. 40-41).

2. It is also surprising that the authors claim in their paper “that data normalisation has no impact on the results” – which begs the question of why include this normalization at all if there are no practical benefits.

As written by this Reviewer in point 2 of the previous round “*normalisation has the advantage of being able to use similar thresholds across images or overlap clusters of cell phenotypes*”.

In practical terms, this means that, after normalisation, pixel intensity values are comparable across samples allowing the user to apply the same marker thresholds. This greatly simplifies and speeds up the analysis configuration. This is explained in the text, where we also explicitly state that normalisation can be skipped and replaced by identifying image-specific thresholds, in case the user prefers to do so (p. 6).

For point 5 of reviewer 2 comments:

3. The updates provided do not address the issues raised. As explained previously, it is not statistically correct to combine cells across different tissue sections and then perform statistical tests (regardless of the filtering the authors have done). This is because cells in an image are not independent of each other. For example, if you have 2 images, one with 500 cells, and the other with 400 cells, following the procedure set out by the authors, there would be ‘900 samples’ (minus some lost by filtering), but in reality, there are only 2 samples. Therefore, their procedure is inadequate and artificially inflates the sample size. Additionally, the use of permutation tests does not support their claims, as permutation tests just test against complete spatial randomness, and

biological tissues are not ‘random’ – therefore, you will likely always get significant p-values.

We think there is a misunderstanding on how the test was performed: we did not filter out specific cell pairs whose distance was below a given value (8 μ m), as the Reviewer seems to imply. Instead, we considered as biologically relevant only those comparisons where the difference of the median distances between the two histological subtypes was higher than 8 μ m. In other words, this filter took into account the effect size between two groups of samples (17 CLRs and 18 DILs) and not between individual pairs of cells. This is now clarified in p. 26 and 41.

Regarding the use of permutations tests, these are commonly employed in the spatial analysis of tissue images (see a recent review by Wilson et al. ²). Examples of spatial analyses that rely on permutation tests include NeighbouRhood³, HistoCAT++⁴, ImaCyte⁵, and Giotto⁶ (all cited in Table 1).

All these tools perform random resampling of the cell labels within each sample, similar to what we have implemented in SIMPLI. In all these cases, the composition of the tissue and the proportion of cell types reshuffled in the permutations are the same as in the original image. This ensures that the generated expected distributions are representative of the original tissue. This is now further explained in the main text (p. 8) and in the methods (p. 33) and Supplementary Fig. 1b.

On how we have implemented the permutation test in SIMPLI, see also point 4 of R3.

Minor issues:

4. Table I contains many inaccuracies. For example, SPIAT does in fact allow unsupervised cell phenotyping as outlined in their manuscript.

From their manuscript and examining the code, we understand that SPIAT implements a semi-supervised and a fully supervised, but not an unsupervised cell phenotyping.

In the semi-supervised approach, a threshold for each marker is either derived from the shape of the distribution or calculated as the 95th percentile of the distribution.

In the fully supervised approach, a list of preselected thresholds are applied.

Either way, cells phenotypes are assigned based on whether the intensity of the markers are above or below these thresholds.

Based on this, we believe it is correct to categorise SPIAT’s cell phenotyping method as “preselected” in Table 1.

Reviewer n. 3

We thank the authors for responding in detail to the points raised in our initial round. We would like to emphasize that the point of addressing the comments below is not to demonstrate the pipeline the authors have developed is perfect in every way, but to better characterize its strengths and weaknesses. We feel it is important to understand what the boundary conditions using SIMPLI are given specific data types and programmatic assumptions that are being made. Since a big part of the value is to unify and simplify image analysis, the end users that will most significantly benefit are groups with less experience in dealing with these datasets. Consequently, it is important for the authors to articulate where SIMPLI should perform well and in what

scenarios it is more prone to erroneous results. With this in mind, we have listed below the areas where we have remaining concerns:

1- The authors stated that they do not want to add MCMICRO to Table 1 because it is not yet peer reviewed. We agree that the fact a study has not yet been published is a reasonable counterpoint to the novelty concerns raised by reviewer 1. However, given that preprints are citeable, we do not feel that it justifies exclusion from surveys of available tools, as many methods (especially computational ones) are available via preprint, and widely used, well in advance of final publication

In the last months there have been several preprints describing new methods for high dimensional imaging data analysis. We have decided not to benchmark SIMPLI against any of the unpublished methods because they are likely to change substantially in the course of the revision process. This is what happened to SIMPLI, which improved thanks to the helpful comments of all reviewers.

We acknowledge that non peer reviewed methods are often used prior to publication. In our opinion, this is a risky choice particularly if the users are not experts in the field or in fact the developers of the method. The reasons for a more cautious approach are exactly the ones exposed by this reviewer in their initial comment: it is important to know the strengths and weaknesses of a method before using it. This can be achieved only after a full peer reviewed process.

2- We share the concerns of reviewer 2 about the normalization technique the authors are proposing. Tissue-specific and platform-specific differences can influence which normalization techniques are used. The fact that the authors have performed channel-specific normalization does not mean that these issues have disappeared. For example, tissues or channels with particularly low signal-to-noise ratios may not work well with the same percentile cutoff used for the rest of a dataset. Furthermore, this cutoff may be different across different image platforms.

As we have now clarified in the revised manuscript (p. 6) and in the GitHub documentation, the normalisation step is optional and can be skipped by the user, for example in the case described by the reviewer (*i. e.* when tissues or channels show particularly low signal-to-noise ratios).

To further assess the effects of normalisation, we compared the analyses of the four case studies reported in the paper done with SIMPLI's normalisation and with alternative approaches:

- IgA⁺ pixel analysis from IMC data: We have compared IgA distribution with and without normalisation showing a strong linear correlation (Supplementary Fig. 2c). This suggests that normalisation does not affect the results. This has been described in the Results (p. 11) and Methods (p. 36).
- Relative proportion of cell types from IMC data: We show that the proportions of T cells, B cells, macrophages, dendritic cells and epithelial cells as obtained after SIMPLI's normalisation (Fig. 3c) are comparable to those obtained by applying HistoCAT's Z-score normalisation (Supplementary Fig. 3b). This is now commented in the Results (p. 15).
- Relative proportions of cell types from mIF data: we obtained similar proportion of PD1⁺CD8⁺ cells and PDL1⁺ cells after applying SIMPLI normalisation (Fig. 4d) or the Inform tissue analysis software¹ distributed with the Vectra Polaris automated

quantitative pathology imaging system (Akoya Biosciences) (Fig. 4e). This is now commented in the Results (p. 20).

- Relative proportion of cell populations from CODEX data: We started from the raw CODEX data and, after data normalisation, we obtained similar proportions of immune cells as in the original paper (Fig. 5b). This is discussed in the Results (p. 25) and Methods (p. 40-41).

3- This is related to our point about benchmarking, as the majority of our concerns centered around preprocessing. We are not asking the authors to evaluate every aspect of their pipeline, since as they correctly note, it encompasses many previously validated individual tools. However, given that different imaging platforms often perform different normalization schemes, the authors' claims that their approach can be used to uniformly process data from all of them merits a more detailed evaluation, as this is a departure from the way some of these data are usually analyzed. In particular, the authors should demonstrate across all of the imaging platforms included that the use of their normalization scheme yields equivalent results to the normalizations commonly used for those data types. This is particularly important since there is no data presented from three of the technologies the authors say their method works on (mIHC, MIBI, and spatial transcriptomics).

We have described four case studies where we apply SIMPLI to IMC, CODEX and mIF data. These experiments differ in terms of resolution, sample size and number of markers (Table 2). We have now provided evidence that, in all cases, the normalisation step as implemented in SIMPLI has no impact on the results.

We feel that adding three more case studies will extend the manuscript length massively worsening its readability, given that we are already approaching the word limits. Although we do not envisage reasons why SIMPLI should under-perform in the case of mIHC, MIBI or spatial transcriptomic data, we acknowledge in the text that the normalisation process may not be directly comparable with the most commonly employed normalisation approaches in these techniques (p. 6).

4- The permutation test analysis provides convincing support for the results of the spatial analysis, but it would be helpful to include it in the SIMPLI pipeline to allow users to assess the significance of their results

This is an excellent suggestion and have now included the possibility to perform a permutation analysis within SIMPLI's spatial pipeline. This is explained in the GitHub documentation as well as in the Results (p. 8), Methods (p. 33) and Supplementary Fig. 1B.

5- The authors apply a Gaussian filter to the IgA images for pixel analysis. It is not clear why this threshold was chosen and if this is a tunable parameter in SIMPLI.

The option to apply a Gaussian filter is part of the CellProfiler pipeline implemented in SIMPLI. This pipeline offers a number of options that can be fully customised and fine-tuned by the user, reflecting their experimental goals and type of analysed images. This is now further clarified in the Results (p. 7), Methods (p. 31) and in the GitHub documentation.

Regarding the IgA analysis discussed in the manuscript, we decided to apply the Gaussian filter after inspection of the tissue slides which revealed a few spurious high intensity pixels that would have biased the downstream analysis. The only tunable parameter associated with the Gaussian filter is the kernel size, which we set to a 1.5 pixel radius, as this is approximately the size of the artefacts that we wanted to remove. This explained in the Methods (p. 36).

References

1. Kramer, A. S. *et al.* InForm software: a semi-automated research tool to identify presumptive human hepatic progenitor cells, and other histological features of pathological significance. *Scientific Reports* **8**, 3418 (2018).
2. Wilson, C. M. *et al.* Challenges and Opportunities in the Statistical Analysis of Multiplex Immunofluorescence Data. *Cancers* **13**(2021).
3. neighbouRhood. <https://github.com/BodenmillerGroup/neighbouRhood> (2019).
4. Schapiro, D. *et al.* histoCAT: analysis of cell phenotypes and interactions in multiplex image cytometry data. *Nature Methods* **14**, 873-876 (2017).
5. Somarakis, A. , Unen, V. V. , Koning, F. , Lelieveldt, B. & Höllt, T. ImaCytE: Visual Exploration of Cellular Micro-Environments for Imaging Mass Cytometry Data. *IEEE Transactions on Visualization and Computer Graphics* **27**, 98-110 (2021).
6. Dries, R. *et al.* Giotto: a toolbox for integrative analysis and visualization of spatial expression data. *Genome Biology* **22**, 78 (2021).

Reviewers' Comments:

Reviewer #3:

Remarks to the Author:

The authors have addressed all of our comments.